# Isolating hydrogen in hexagonal boron nitride bubbles by a plasma treatment

Li He[1,2,3], Huishan Wang[2,3,4], Lingxiu Chen [2,3], Xiujun Wang[2,3,4], Hong Xie[2,3], Chengxin Jiang[2,3,5], Chen Li[6,11], Kenan Elibol[7,12,13], Jannik Meyer [7,14], Kenji Watanabe [8], Takashi Taniguchi[8], Zhangting Wu[9], Wenhui Wang[9], Zhenhua Ni[9], Xiangshui Miao[1], Chi Zhang[1], Daoli Zhang [1], Haomin Wang [2,3,10] & Xiaoming Xie[2,3,5]

Atomically thin hexagonal boron nitride (h-BN) is often regarded as an elastic film that is impermeable to gases. The high stabilities in thermal and chemical properties allow h-BN to serve as a gas barrier under extreme conditions. Here, we demonstrate the isolation of hydrogen in bubbles of h-BN via plasma treatment. Detailed characterizations reveal that the substrates do not show chemical change after treatment. The bubbles are found to withstand thermal treatment in air, even at 800 °C. Scanning transmission electron microscopy investigation shows that the h-BN multilayer has a unique aligned porous stacking nature, which is essential for the character of being transparent to atomic hydrogen but impermeable to hydrogen molecules. In addition, we successfully demonstrated the extraction of hydrogen gases from gaseous compounds or mixtures containing hydrogen element. The successful production of hydrogen bubbles on h-BN flakes has potential for further application in nano/micro-electromechanical systems and hydrogen storage.

[1] School of Optical and Electronic Information, Huazhong University of Science and Technology, 430074 Wuhan, China. [2] State Key Laboratory of Functional Materials for Informatics, Shanghai Institute of Microsystem and Information Technology, Chinese Academy of Sciences, 865 Changning Road, 200050 Shanghai, China. [3] CAS Center for Excellence in Superconducting Electronics (CENSE), 200050 Shanghai, China. [4] Graduate University of the Chinese Academy of Sciences, 100049 Beijing, China. [5] School of Physical Science and Technology, ShanghaiTech University, 319 Yueyang Road, 200031 Shanghai, China. [6] Department of Lithospheric Research, University of Vienna, Althanstraße 14, 1090 Vienna, Austria. [7] Faculty of Physics, University of Vienna, Boltzmanngasse 5, 1090 Wien, Austria. [8] National Institute for Materials Science, 1-1 Namiki, Tsukuba 305-0044, Japan. [9] Department of Physics, Southeast University, 211189 Nanjing, China. [10] Center of Materials Science and Optoelectronics Engineering, University of Chinese Academy of Sciences, 100049 Beijing, P.R. China. [11] Present address: Electron Microscopy for Materials Research (EMAT), University Antwerpen, Groenenborgerlaan 171, 2020 Antwerpen, Belgium. [12] Present address: School of Chemistry, Trinity College Dublin, Dublin 2, Ireland. [13] Present address: Advanced Microscopy Laboratory, Centre for Research on Adaptive Nanostructures and Nanodevices, Dublin 2, Ireland. [14] Present address: Institute for Applied Physics and Natural and Medical Sciences Institute, University of Tübingen, Tübingen D-72076, Germany. Correspondence and requests for materials should be addressed to D.Z. (email: zhang_daoli@hust.edu.cn) or to H.W. (email: hmwang@mail.sim.ac.cn)

Hexagonal boron nitride (abbreviated as *h*-BN in the following) is a remarkable two-dimensional mesh consisting of alternating $sp^2$-bonded boron and nitrogen atoms[1]. As a wide bandgap insulator[2], *h*-BN is regarded as an ideal substrate and tunneling barrier for graphene devices because of its atomically smooth surface free of dangling bonds and charge traps[3]. In addition, the exceptional thermal and chemical stabilities of *h*-BN enable it to function as an antioxidation layer, even at temperatures as high as 1100 °C[4]. Monolayer *h*-BN sheets remains stable up to 800 °C in air, whereas graphene oxidizes at temperature above 500 °C[5]. This makes *h*-BN a top choice for ultrathin antioxidization coatings. In addition to the potential for application in electronics and corrosion-proof coatings, *h*-BN holds promise for selective membrane applications under extreme conditions due to its ultimate thinness, flexibility, and mechanical strength[4,6–8].

Similar to graphene[9], *h*-BN is elastic and impermeable to all gases. High-quality *h*-BN sheets have a Young's modulus of ~0.85 TPa and an excellent fracture strength of ~70 GPa[8]. The sheets can withstand stretching of up to 20%[8]. This feature makes *h*-BN a superb material for application in microelectromechanical systems (MEMSs). A perfect *h*-BN crystal, even when just a monolayer thick, is typically impermeable to most atoms and molecules under ambient conditions. Only accelerated hydrogen atoms possess sufficient kinetic energy to penetrate through *h*-BN planes without damaging the lattices[10,11]. These properties allow bubbles with various shapes to be produced on *h*-BN[12]. Recently, several reports have described the trapping of gases by graphene[12–16]. By forming bubble structures, the adhesion energy between graphene sheets and substrate can be precisely measured[17,18]. A voltage-controlled graphene micro-lens with variable focuses[19] and a laser-driven nano-engine[20] have been realized, as the curvature and shape of the bubbles can be controlled to a great extent by variation in the voltage. Although *h*-BN has a similar honeycomb lattice to graphene, little investigation of *h*-BN bubbles for gas isolation has been reported. Compared to graphene, *h*-BN is insulating and more chemically stable, which may greatly extend its application in extreme conditions.

There have been several reports about plasma treatment on 2D materials. Most of them are about plasma applications in surface etching[21,22], low-temperature growth[23,24] and surface hydrogenation[25]. In this work, we demonstrate the isolation of hydrogen in *h*-BN bubbles on the microscale via plasma treatment. The dimension of the bubbles can be regulated by altering the treatment conditions. The diameter of the bubbles ranges from tens of nanometers to several micrometers. The masses of $H_2$ molecules inside the bubbles are believed to have been converted to atomic hydrogen by the plasma, which permeate through the *h*-BN plane and is then captured inside the *h*-BN interlayers. Re-formation of $H_2$ molecules deforms the *h*-BN structure to form bubbles, as the $H_2$ molecules cannot escape from the *h*-BN net. Our demonstration provides an effective method for fabricating bubbles on *h*-BN and may also be an approach for extracting/storing hydrogen in *h*-BN.

## Results

**Bubble formation on *h*-BN**. Figure 1a shows a schematic illustration of the experimental design for the bombardment of the *h*-BN surface by plasma in different gaseous environments. As shown in Fig. 1, three different gases (Ar, $O_2$, and $H_2$) were fed into the chamber to examine the ability of different elements to penetrate *h*-BN. The flow rate was kept at 3 sccm, while the pressure was kept at ~3 Pa. Plasmas were then generated by a RF generator with a power of 100 W. All *h*-BN flakes were treated with the plasma at 350 °C for 150 min. Atomic force microscopy (AFM) images of the *h*-BN flakes were taken after the plasma treatment and are given in Fig. 1b–d (the corresponding optical images are given in Supplementary Fig. 1). As shown in Fig. 1b–d, the *h*-BN flakes treated with argon or oxygen plasma appear to be intact, while a large amount of bubbles appeared on the *h*-BN surface that was treated with the hydrogen plasma (H-plasma). A series of experiments including structural characterization by transmission electron microscopy (TEM) (see Supplementary Figs. 2 and 3), chemical characterization by X-ray photoelectron spectroscopy (XPS) (see Supplementary Figs. 4 and 5 and Supplementary Table 1), Energy-dispersive X-ray spectroscopy (EDX) (see Supplementary Figs. 6 and 7), Fourier Transform infrared spectroscopy (FTIR) (see Supplementary Fig. 8) and mass spectrometry (see Supplementary Figs. 9 and 10) have been conducted to verify the *h*-BN

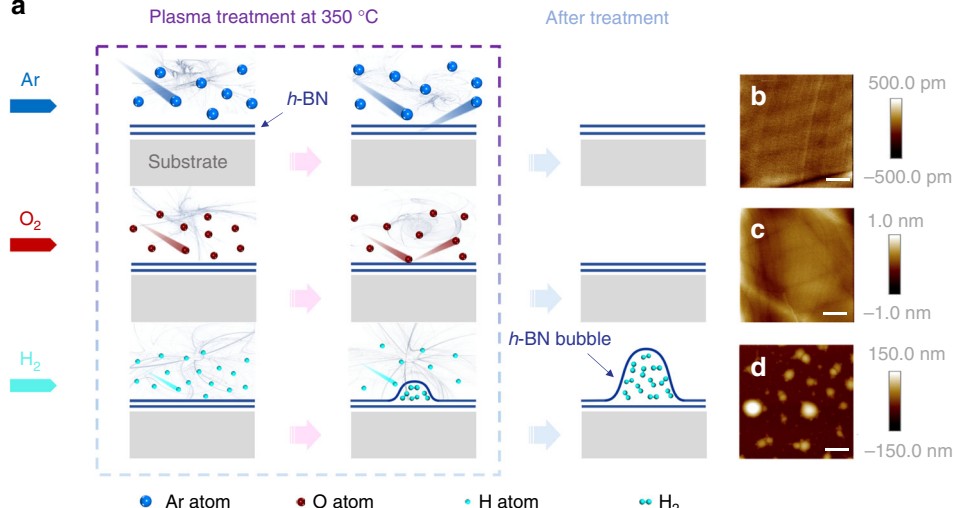

**Fig. 1** Production of bubbles on *h*-BN flakes via plasma treatment. **a** Schematic depicting the plasma treatment of *h*-BN flakes in different atmospheres. All AFM height images of the *h*-BN flakes were taken after plasma treatment in an atmosphere of **b** argon, **c** oxygen or **d** hydrogen. The *h*-BN flakes appear to remain intact after the argon and oxygen plasma treatments, while obvious bubbles are observed on the *h*-BN surface after treatment with the H-plasma. All samples were obtained under similar conditions. Scale bars in (**b**–**d**) are 4 μm

bubble structure intact and the presence of hydrogen gases inside the bubbles.

To understand the mechanism behind the interesting bubble formation is important. It is well known that gaseous molecules subjected to a strong electromagnetic field decompose into a plasma state, producing masses of charged ions whose particle radii are considerably decreased relative to the original atoms. The hexagonal honeycomb lattice of $h$-BN possesses a lattice constant ($a_{h\text{-BN}}$) of ~250.4 pm[26]. A hydrogen atom has a diameter of ~240 pm (with a van der Waals radius of ~120 pm[27,28]), which is less than the value of $a_{h\text{-BN}}$ and could pass through the valence electrons concentrated around the BN atoms. Moreover, large amounts of protons, which have a diameter of ~1.68 fm[29,30], are generated in the plasma state. These extremely small particles can easily penetrate the BN. The accelerated energetic protons and electrons could then recombine to form hydrogen molecules. These hydrogen molecules are not easily able to pass through the $h$-BN honeycomb due to their large kinetic diameter of ~289 pm[31], which is substantially larger than $a_{h\text{-BN}}$. Thus, the pressure from the trapped gas causes bubbles to form on the surface of the $h$-BN flakes. In contrast, the atomic diameters of both argon (~376 pm) and oxygen (~304 pm) (with the van der Waals radii of ~188 pm and ~152 pm, respectively[27,28]) are considerably larger than $a_{h\text{-BN}}$, and as a result, they cannot penetrate the $h$-BN honeycomb barrier.

The mechanism of unimpeded penetration of atomic hydrogen into the $h$-BN interlayers can also be ascribed to two additional reasons. The first one is the polarization of the electron density distribution in this kind of 2D material. Recent experiments have demonstrated that $h$-BN possesses the best porosity among 2D materials[11]. The concentration of valence electrons around the N atoms results in strong polarization of the BN bonds, making

$h$-BN much more porous than graphene and MoS$_2$. The second reason is the stacking order. It has previously been confirmed in the former work that bi- and tri-layered graphene are also nonconductive to protons due to their staggered lattices (AB stacked), in which the electron cloud of the first layer clogs the pores of the second[11]. In contrast, multilayered $h$-BN exhibit excellent proton conductivity probably due to its favoring the AA′ stacking structure, which has been found to be the most stable one among the five possible stacking structures (AA′/AA/AB/AB′/A′B) in recent calculations[32]. The aligned pores in the out-of-plane direction also allow protons to move unimpededly throughout the space. This characteristic structure of multilayer $h$-BN could facilitate its use in hydrogen storage applications.

To study the stacking order of the $h$-BN multilayer, a scanning transmission electron microscopy (STEM) investigation was carried out. The $h$-BN flakes are firstly exfoliated onto a quartz substrate, and then annealed to remove the residue of resist in an oxygen flow at 800 °C. After cooling down to room temperature, a group of wrinkles are found on the surfaces. The wrinkles, which are found along the armchair direction of $h$-BN crystals[33,34], help us determine the cutting direction of the focused ion beam. The specimen obtained for the STEM investigation is similar to that shown in Supplementary Fig. 3a.

Figure 2a illustrates that the incident electron beam (e-beam) for the STEM investigation is parallel with the [11$\bar{2}$0] crystallographic axis of $h$-BN. The cross-sectional schematic of the armchair edge-on view of $h$-BN multilayer is presented in Fig. 2b. Figure 2c is a Wiener filtered STEM medium angel annular-dark-field (MAADF) image showing the edge-on view configuration of $h$-BN multilayer, which reveals the layer stacking of the $h$-BN. The intensity profiles along two selected lines (red/blue) in Fig. 2c are showed in Fig. 2d, exhibiting apparent synchronism of the peak

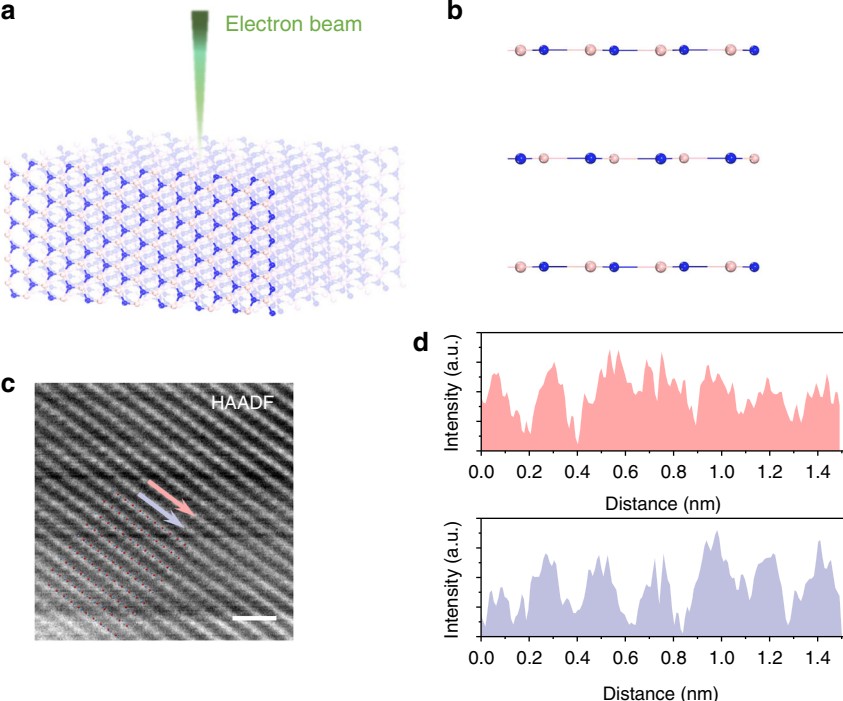

**Fig. 2** STEM image showing the edge-on view configuration of an $h$-BN multilayer. **a** Schematic of a STEM investigation of the $h$-BN multilayer, the incident electron beam was parallel with its [11$\bar{2}$0] crystallographic axis (the zigzag edge of the $h$-BN layer). **b** Cross-sectional schematic of the armchair edge-on view of $h$-BN multilayer. The red atoms represent boron atoms while the blue ones represent nitrogen atoms. **c** STEM-MAADF image, postprocessed by a Wiener Filter, showing the edge-on view configuration γ of an $h$-BN multilayer. Scale bar in (**c**) indicates 1 nm. **d** Profiles of image intensity along the red and blue arrows marked in (**c**), showing the stacking sequence in $h$-BN. Their average peak-to-peak distances are ~2.26 Å, which are in agreement with the lattice spacing of $h$-BN. The synchronism of the peak intensity along the arrows demonstrates the stacking structure of $h$-BN must be either AA′ or AA

positions. Besides, the average peak-to-peak distances measured in red and blue profiles are both ~2.26 Å, which are close to 2.166 Å expected in *h*-BN. The minor deviation of the spacing in STEM measurement might be due to the limitation of the scale calibration of the instrument. This result shows that the stacking order of our *h*-BN is either AA′ or AA in our *h*-BN multilayer, which is in line with earlier predictions. Beside the cross-sectional STEM image, we also carried out a plan-view STEM investigation (shown in Supplementary Fig. 11) of another *h*-BN multilayer. The image confirms that the *h*-BN multilayer we fabricated is in AA′ stacking. These images provide direct evidence that the multilayered planes in *h*-BN are not staggered so that the electron cloud of each layer does not block the pores of the successive layer. As such, atomic hydrogen could tunnel through multilayered *h*-BN.

**Control on size of *h*-BN bubbles**. Both the duration and the environmental temperature of plasma treatment have prominent

effects on the density and sizes of bubbles, which provide a feasible way to fabricate *h*-BN bubbles with controllable dimensions. To understand the dependence of the bubble dimensions on the duration of the H-plasma treatment, we varied the treatment duration while holding the other conditions constant (100 W RF, 350 °C, 3 sccm $H_2$, ~3 Pa) and examined the variations in the dimensions of the generated bubbles. The results of this experiment are given in Fig. 3. As shown in Fig. 3a, the *h*-BN samples obtained in different duration of H-plasma treatment possessed bubbles with obviously different densities and size distributions. The bubbles treated for 90 min were generally smaller than 100 nm in diameter and had a higher density in the distribution, while plasma treatment for 150 min led to a lower bubble density and a larger bubble size with diameters greater than 3 μm. The dependence of diameter distribution on treatment duration is plotted in Fig. 3b. Extended plasma treatment favors the formation of larger bubbles. An H-plasma treatment for 500 min could

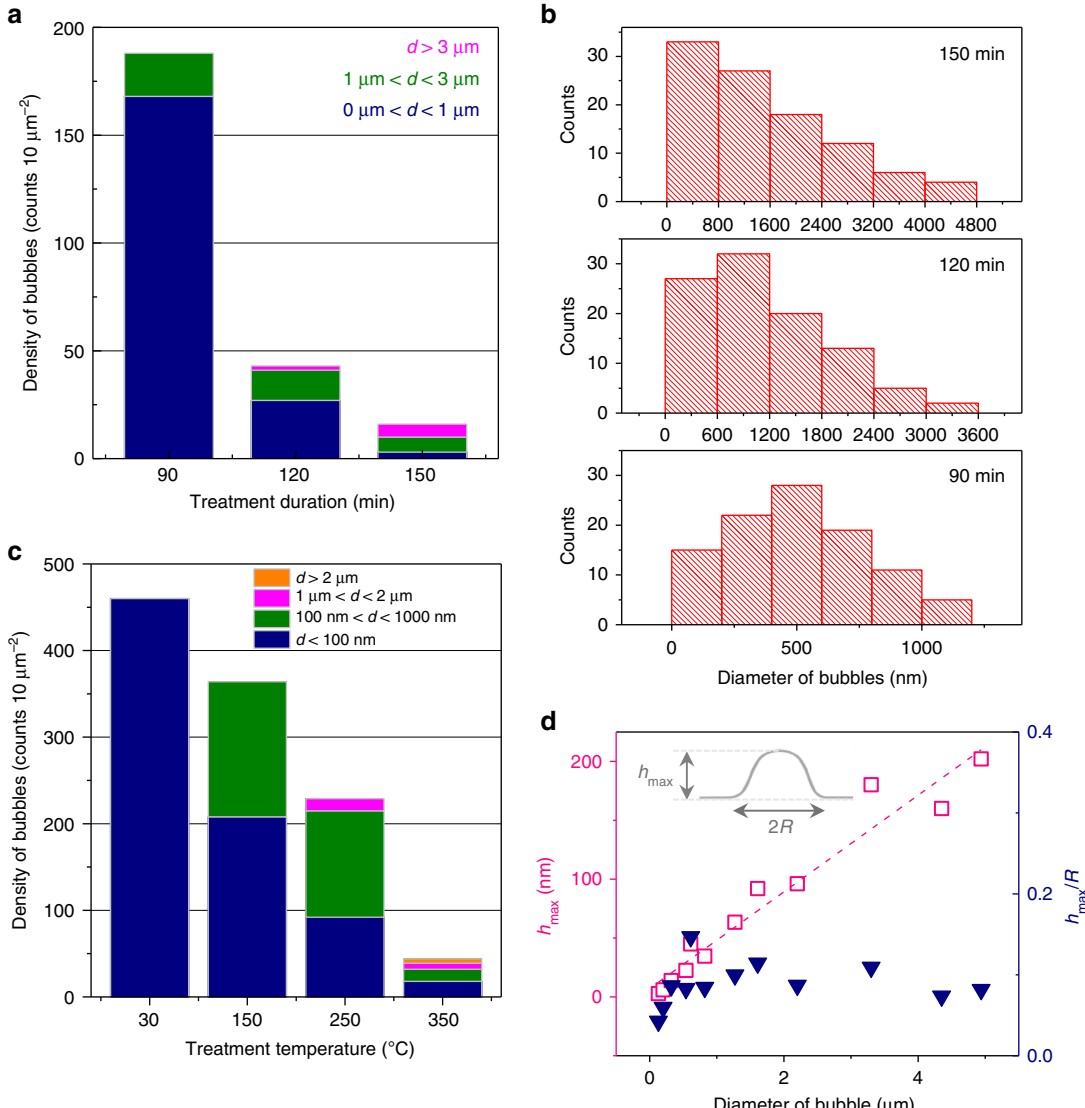

**Fig. 3** Influence of the plasma treatment duration and sample temperature on the dimensions of the bubbles. **a** Density and relative fractions of *h*-BN bubbles within three diameter ranges (0–1 μm, 1–3 μm, and >3 μm) produced by different H-plasma treatment durations; all samples were treated under the typical experimental conditions (100 W RF, 350 °C, 3 sccm $H_2$, ~3 Pa). **b** Dimension distributions of the *h*-BN bubbles produced after H-plasma treatment for 90, 120, and 150 min. **c** Density and relative fractions of *h*-BN bubbles within four diameter ranges (<100 nm, 0.1–1 μm, 1–2 μm, and >2 μm) produced from treatment at different sample temperatures (30, 150, 250 and 350 °C); the other conditions were not changed (100 W RF, 120 minutes, 3 sccm $H_2$, ~3 Pa). **d** $h_{max}$ of the bubbles produced by the H-plasma and the aspect ratio of the bubbles with respect to the bubble diameter. The inset shows an illustration of $h_{max}$ and $R$

fabricate larger bubbles. As shown in Supplementary Fig. 12a–c, bubbles with diameters of ~20 μm were obtained on $h$-BN.

In addition to the treatment duration, the environmental/sample temperature is another key factor that influences the dimensions of the bubbles. Thus, the sample temperature was also adjusted while keeping the other conditions constant (100 W RF, 120 min, 3 sccm $H_2$, ~3 Pa) to investigate the variations in the bubble dimensions. Figure 3c presents the density and diameter fractions of the $h$-BN bubbles for different sample temperatures. Almost all of the bubbles produced at room temperature (30 °C) were <100 nm in diameter, and the relative fraction of large bubbles gradually increased when the sample temperature was elevated from 30 to 350 °C. Meanwhile, the density of the bubbles dramatically decreased with increasing sample temperature. Furthermore, the bubble shown at the bottom-left of Supplementary Fig. 13d formed by merging three individual bubbles. The observation reveals that the bubbles can migrate and merge at higher sample temperature. Similar phenomena were also observed in Fig. 3a and Supplementary Fig. 14. The hydrogen molecules inside smaller bubbles, normally possessing higher inner pressure than relatively large bubbles, move more easily at higher sample temperatures in-between $h$-BN interlayers[15]. Hydrogen gas in small bubbles is more likely to pass through the inter-planar channels and combine to form a larger bubble with a decreased inner pressure. Such combination leads to a dramatic decrease in bubble density at high sample temperatures. Detailed investigation in the stability of bubbles on $h$-BN is given in Supplementary Figs. 15–20.

We then further explored the relationship between the height/aspect ratio of the bubbles and their diameter. The maximum height, $h_{max}$, and the quotient of $h_{max}$ to the radius $R$ (called the aspect ratio) of each bubble are taken from all observed $h$-BN bubbles by AFM. The results are presented in Fig. 3d. It is found that $h_{max}$ approximately increases linearly with bubble diameter, while the aspect ratio remains almost unchanged.

**Mechanisms of the plasma-driven effect**. In order to understand the mechanism behind the plasma process, exploring hydrogen plasma itself is essential[35]. We carried out modeling of hydrogen plasma in a tube using COMSOL Multiphysics. The spatial distribution of different parameters in the tube are plotted in Supplementary Fig. 21. As shown in Supplementary Fig. 21e, f, k, l, the ion density at the sample area decreases while the thermal velocity increases, when the sample was heated from room temperature to 350 °C. The results indicate that protons in H-plasma can be injected much faster into $h$-BN interlayers at a higher environmental temperature with comparatively lower spatial density. This simulation exhibits the variation tendency of dimension/density when elevating the environmental temperature (heated by the furnace) of the sample (as shown in Fig. 3c). The furnace temperature (Supplementary Fig. 21a, b), electron density (Supplementary Fig. 21c, d), electron temperature (Supplementary Fig. 21g, h), and pressure (Supplementary Fig. 21i, j) are also given to help in giving images shown plasma-driven kinetics.

It is noticed that the electron density (Supplementary Fig. 21c, d) is two orders in magnitude lower than the ion density (Supplementary Fig. 21e, f) at the sample position according to simulation. This phenomenon indicates that the quasi-neutrality of plasma at the sample position is broken. According to the setup showed in Supplementary Fig. 22a–d, $h$-BN sample was placed 30 cm away from the RF coil (the core region of the plasma). Along the axial direction of tube furnace, there is a steep electron gradient which leads to the unbalance between electron density and ion density, and thus, the quasi-neutrality of plasma is not sustained.

To understand this phenomenon further, we then carried out modeling in Debye length when the furnace was set at 350 °C in Supplementary Fig. 21q. Normally, Debye length depends on density of ion number and environmental temperature. Debye length increases when ion number density decreases or when the environmental temperature increases. It is also found that the Debye length, which is much higher than the diameter of the tube (0.042 m), reached its maximum of ~0.5 m at the sample position. It indicates the destruction of quasi-neutrality at the sample position. This also interprets why the electron density is two orders lower than the ion density at the sample position.

Besides the sample temperature and treatment duration, the position where the $h$-BN sample was placed also brings different size/distribution of bubble on $h$-BN. The schematic of experimental setup as well as the results of bubble distribution are exhibited in Supplementary Figs. 23 and 24, respectively. It is obvious that bubbles in the plasma core area are small but dense. while bubbles at other area show relatively large but thin. The electric potential difference (Supplementary Fig. 21m, n) and the ion number density distribution (Supplementary Fig. 21o, p) are also simulated by COMSOL Multiphysics to analyze the bubble distributions in the supporting information. Obviously, the formation of bubbles is a plasma-driven effect. The size and distribution of bubbles on $h$-BN are generally determined by the comprehensive influence of ion density, Debye length and potential gradient at the position where the $h$-BN samples are placed.

**Low-temperature AFM measurement**. To determine the type of gases inside the $h$-BN bubbles, low-temperature AFM measurements were carried out to explore their temperature evolution. The experiment was inspired by a very recent literature about bubble structure on bulk transition metal dichalcogenides[36]. Details about measurement are given in the Methods section. Figure 4a shows an optical image of bubbles on an $h$-BN flake. After transferring it into a vacuum chamber equipped with an AFM, the sample was cooled down gradually. Figure 4b shows topographic AFM images of the same area measured at 34 and 33 K respectively. It is clearly shown that the bubbles deflate at 33 K while they are inflated with gases at 34 K. As shown in Fig. 4c, the height profiles of line-scans across a bubble (indicated by dashed lines in Fig. 4b) clearly show that the bubble remains at ~34 K and disappears at ~33 K. The deflating/swelling processes were reproducible via cooling/heating the sample. The highest temperature at which the bubbles become flat was recognized as the transition temperature ($T_{transition}$). We measured a total of 58 bubbles. Three other typical samples are given in Supplementary Figs. 25–27. The transition temperature of each bubble was recorded when they become flat during cooling. The number of bubbles which have the same $T_{transition}$ is counted. The histogram of $T_{transition}$ as function of temperature at an interval of 1 K is plotted in Fig. 4d. As shown in Fig. 4d, $T_{transition}$ of all the bubbles measured can be described as a Gaussian distribution, and exhibits an average value equal to 33.2 ± 3.9 K. This value is very close to $T_{transition}$ of $H_2$ (33.18 K) among all possible gases (see Supplementary Table 2). The result strongly indicates that the gases inside the bubbles are indeed hydrogen molecules.

**Determining the penetration depth of atomic hydrogen in the $h$-BN multilayer**. Increasing the power of the RF source may lead to bursting of the bubbles, which leaves many pits on the $h$-BN surface. These pits make it possible to study the penetration depth of atomic hydrogen. Figure 5a shows an optical micrograph of an $h$-BN flake treated for 180 min at 350 °C with an RF power of 400 W. Numerous pits are observed on the $h$-BN surface. Figure 5b, c depict the AFM image of a deep pit highlighted by

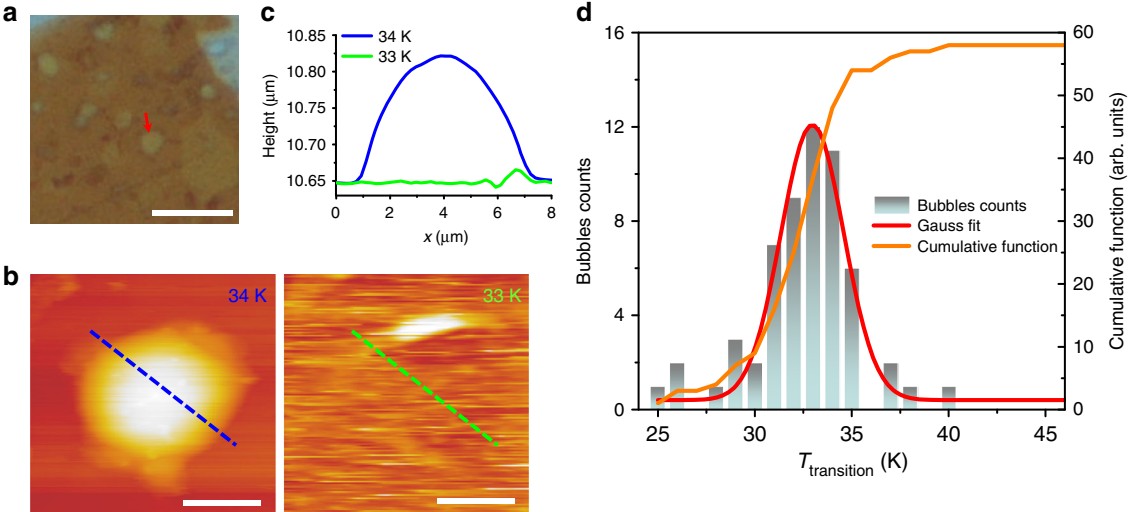

**Fig. 4** Swelling and deflating processes of the $h$-BN bubbles containing hydrogen. **a** An optical image of bubbles on an $h$-BN flake, taken under ambient condition, scale bar: 20 μm. **b** Topographic AFM image of a bubble pointed-out by an arrow in (**a**) was measured at 34 and 33 K respectively, scale bars: 3 μm. **c** The height profiles of line-scan at the same place (indicated by dashed lines in (**b**)) where the bubble remains at ~34 K and disappears at ~33 K. **d** Histogram of the transition temperature ($T_{transition}$) at which bubbles collapse. The red line is a Gaussian fit to the data. The yellow line is the histogram cumulative function (right axis)

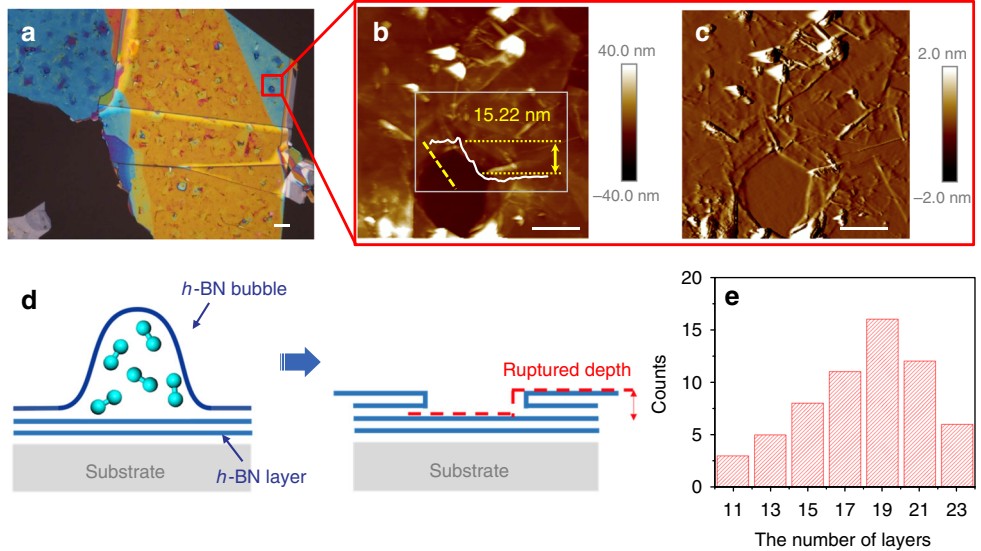

**Fig. 5** Penetration depth of atomic hydrogen into $h$-BN multilayers. **a** Optical image of an $h$-BN flake on which the bubbles generated by a 400 W H-plasma treatment ruptured. $h$-BN flaps peeled off the adhered macroscopic films, and the multilayer $h$-BN sheets tore and then folded. **b** AFM height and **c** amplitude error images taken of the red area shown in (**a**). The bottom of the pit is atomically flat, and no bubble filled with hydrogen molecules was found. The insert in (**b**) is the AFM depth profile along the yellow dashed line. The deviation in the height is approximately 15.2 nm, indicating that ~23 atomic layers were ruptured and folded over the top layer of $h$-BN. The scale bars in (**a**–**c**) represent 1 μm. **d** Illustration of the ruptured depth for the folding steps depicted in (**b**, **c**) caused by the bursting of the $h$-BN bubble. **e** Depth distribution of penetration into the surface of the $h$-BN flake shown in (**a**)

the red box in Fig. 5a. The AFM height image in Fig. 5b provides depth information about the pit, whose depth is given in the insert. The amplitude error image provided in Fig. 5c shows a clearer picture of the morphology around the pit. The bottom of the pit is smooth, which indicates that no atomic hydrogen penetrated the bottom layer. Furthermore, folds are observed in the $h$-BN flaps around the outside of the pit, as illustrated in Fig. 5d. The inset in Fig. 5b shows that the rupture depth of the folded border step is approximately 15.22 nm, which indicates that ~23 layers of $h$-BN were penetrated by atomic hydrogen in the 400 W treatment. Figure 5e presents the statistics of the penetration depth of atomic hydrogen in the $h$-BN sample shown

in Fig. 5a. The penetration depth of the $h$-BN flake ranges from ~11 to ~23 layers. The histogram shown in Fig. 5e reveals that the 400 W RF power provided the H-plasma with enough kinetic energy to penetrate at most ~23 atomic layers underneath the $h$-BN surface.

**Characterizations of Raman spectroscopy.** Raman spectroscopy was also performed to characterize the $h$-BN bubble. As shown in Fig. 6a, a typical bubble with a round shape was measured. The bubble had a diameter of ~2.36 μm and a height of ~122 nm. The Raman spectrum was measured at three spots from the flat area next to the bubble to the center of the bubble. The results

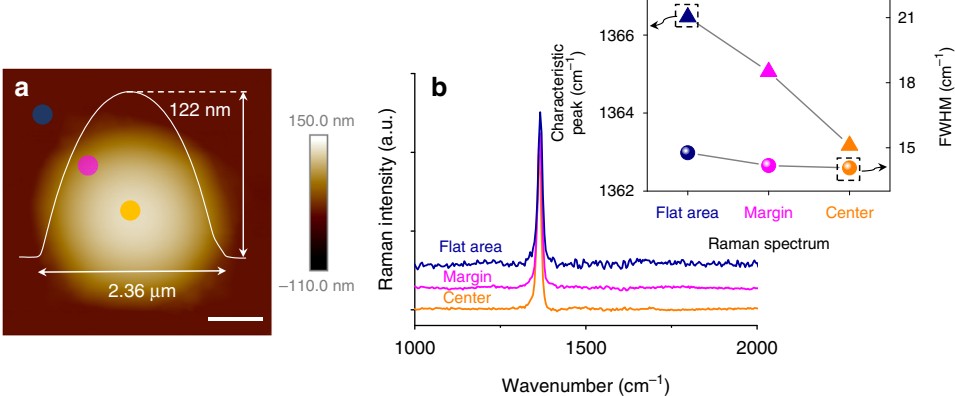

**Fig. 6** Raman spectra taken at different positions of an *h*-BN bubble. **a** AFM image of a typical *h*-BN bubble; the white curve shows the profile across the center of bubble and gives information about its height and diameter. Scale bar, 600 nm. **b** Raman spectra taken at the positions indicated in the AFM image. The inset shows the variation in the $E_{2g}$ peak position and FWHM between positions. A redshift in the $E_{2g}$ peak position from the flat area to the center of the bubble is observed, while the corresponding FWHM is nearly unchanged

are shown in Fig. 6b. The Raman spectra of *h*-BN clearly show only one peak near 1366 cm$^{-1}$, which corresponds to the $E_{2g}$ vibration mode of *h*-BN. As shown in the inset of Fig. 6b, the $E_{2g}$ peak position and full width at half maximum (FWHM) of each spectrum were extracted and plotted as a function of the measurement site. The peak position in the Raman spectrum appeared at the center of the bubble redshifted (~3 cm$^{-1}$) relative to that of the position outside of the bubble, with almost no change in the FWHM. The observed redshift in the peak position is due to the presence of stretching strain on the bubble caused by expansion[37,38]. Further investigation in the temperature dependence of Raman spectra of *h*-BN bubble is given in Supplementary Fig. 28.

**Hydrogen extraction**. The demonstrated fabrication of hydrogen bubbles on *h*-BN via plasma treatment inspired us to consider the possibility of extracting hydrogen from hydrocarbons and mixtures of gases. Three different gases ($C_2H_2$, $CH_4$ and $Ar + H_2$, 5% of $H_2$) were fed into the chamber for plasma treatment under the same experimental conditions (350 °C, 100 W, 120 min, ~3 sccm flow rate, ~3 Pa). Excitingly, all three samples formed bubble structures on the *h*-BN surface after the plasma treatment. The experimental illustration and AFM images of the *h*-BN flakes are given in Fig. 7a–f. Considering that the diameters of carbon and argon atoms are much larger than the lattice constant of *h*-BN, it is likely that only atomic hydrogen could penetrate the *h*-BN layers. The results indicate that hydrogen could be separated from hydrocarbons and from the mixture of argon and hydrogen.

As shown in Fig. 7d–f, the bubbles formed after the ethyne plasma treatment had diameters of predominately <300 nm and a high density, while the bubbles produced by the methane plasma possessed a much larger diameter range from 2 to 4 μm with a much lower density. The exact causes for the phenomena need comprehensive investigation in future. At this moment, the main reason, we believe, could be that the dissociation energy of methane is actually lower than that of the ethyne[39]. When all experiments were carried out under similar conditions, more atomic hydrogen in $CH_4$ plasma can be generated and then more efficiently get into *h*-BN bubbles than the case in ethyne plasma. The plasma treatment with 5% hydrogen in argon also led to the formation of bubbles on the *h*-BN surface but with an extremely low density. To characterize the bubbles further, the plots of $h_{max}$ and the aspect ratio with respect to the bubble diameter are presented in Fig. 7g–i. Similar to the bubbles fabricated by the

plasma of pure hydrogen, in all three plots, $h_{max}$ had a nearly linear increasing relationship with the bubble diameter, while the aspect ratio remained nearly unchanged and thus was independent of the bubble diameter. Interestingly, although the bubbles produced from $CH_4$ had much larger diameters (2–4 μm) than those produced from $C_2H_2$ (<300 nm), they had almost the same $h_{max}$ (<10 nm), resulting in an extremely low aspect ratio ($h_{max}/R$ = ~0.005) for the bubbles produced from methane.

## Discussion

We demonstrate an approach for isolating hydrogen in *h*-BN bubbles by plasma treatment. The dimensions of the bubbles range from nanometers to micrometers and depend on the duration and sample temperature of the plasma treatment. Most of the bubbles undergo minimal changes over 40 weeks. Additionally, the penetration depth of atomic hydrogen reaches approximately 23 *h*-BN layers. The Raman spectra of a bubble reveal that stretching occurs on the bubble surface. Finally, we demonstrate that hydrogen gas can be extracted from hydrocarbons and mixed gases by the plasma treatment and isolated between *h*-BN layers in bubbles. The demonstrated ability to fabricate hydrogen bubbles on *h*-BN flakes might be used to extract hydrogen from hydrocarbon/gas mixtures or to fabricate nano/micro-electromechanical systems.

## Methods

**Fabrication and characterization of hydrogen bubbles on *h*-BN**. The experimental setup is given in Supplementary Fig. 22. First, *h*-BN flakes were produced by micromechanical exfoliation of single-crystal *h*-BN and then deposited on quartz substrates that were previously cleaned with oxygen plasma. The substrates with *h*-BN flakes were loaded into a furnace equipped with an RF generator (13.56 MHz, MTI Corporation) with a tunable power from 100 to 400 W. The sample temperature in the sample chamber could be controlled from room temperature to 1000 °C. An advanced pump (GX100N Dry Pumping System, Edwards) was connected to the chamber to control the flow rate. After the samples were heated to the preset temperature, hydrogen (oxygen or argon) gas with a flow rate of 3 sccm (~3 Pa) was introduced into the chamber, and then the plasma was formed. The plasma treatment normally took 90, 120 or 150 min. The *h*-BN flakes were finally removed from the sample chamber for characterization under an optical microscope (Eclipse LV150, Nikon), an atomic force microscope (Dimension Icon, Bruker), and a Raman spectrometer (532 nm, WITec).

**Low-temperature AFM measurement of the bubbles**. After bubbles were produced on *h*-BN flakes, optical microscope and AFM were used to locate their positions. Subsequently, the substrate with the *h*-BN flakes was transferred into the vacuum chamber of a variable temperature SPM (Scanning probe microscope) system (attoDRY 1100 by Attocube). The chamber was filled with $^4$He in a pressure of 5 mbar for heat exchange. The sample temperature can be controlled in the range from 4 to 300 K. All the AFM scanning was conducted in contact mode.

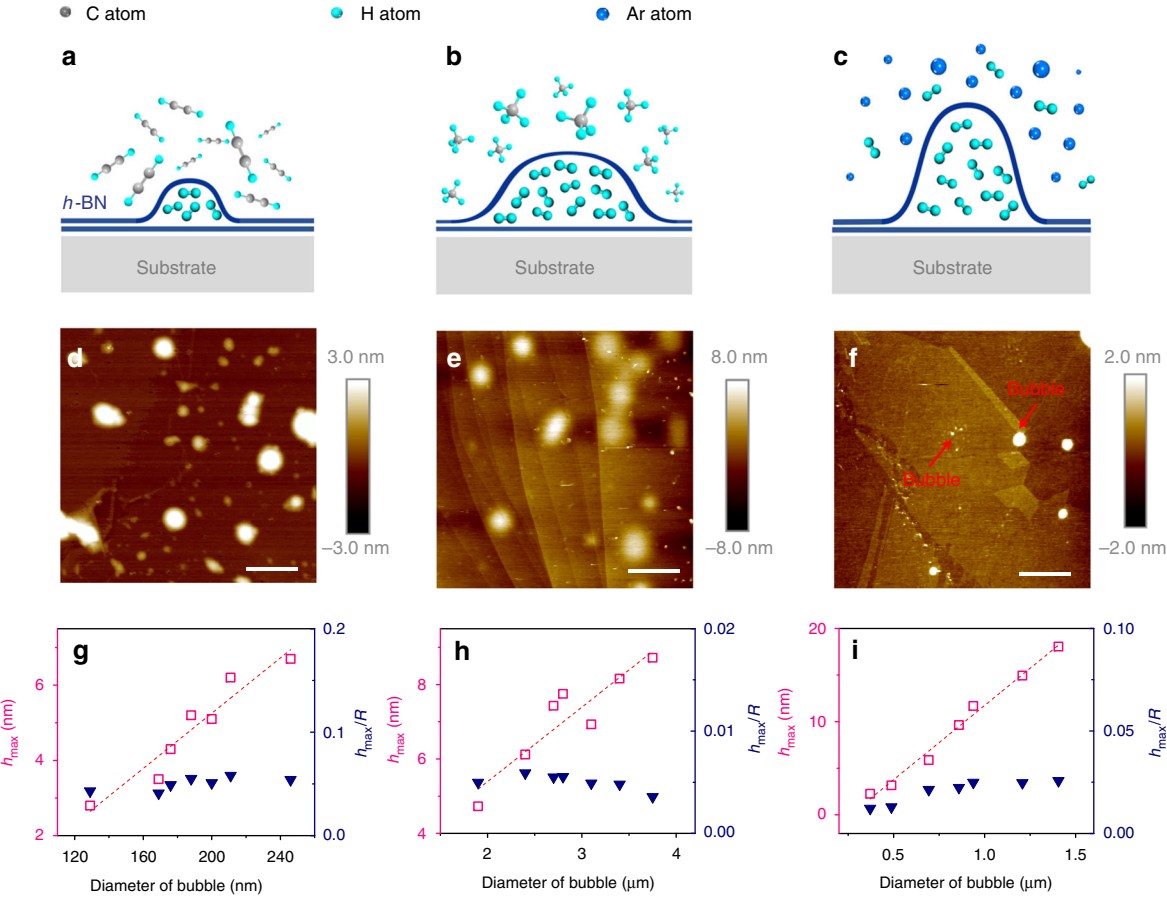

**Fig. 7** Extracting hydrogen from hydrocarbon and mixture gases. **a–c** Schematic illustration of the separation of atomic hydrogen from ethyne ($C_2H_2$), methane ($CH_4$) and a mixture of hydrogen and argon (Ar + $H_2$, 5% $H_2$) by an *h*-BN membrane via plasma treatment. **d–f** AFM height images of the *h*-BN surfaces treated by a typical plasma process (350 °C, 100 W, 120 min, ~3 sccm, ~3 Pa) in an environment of ethyne, methane and a mixture of hydrogen and argon. Scale bars: **d** 600 nm, **e**, **f** 4 μm. **g–i** $h_{max}$ of the bubbles fabricated via the plasma treatment in $C_2H_2$, $CH_4$ and Ar + $H_2$ (5% of $H_2$), respectively, and their aspect ratios as a function of the bubble diameter

In order to minimize the damage to the bubbles, AFM tips with a force constant of 0.02–0.77 N m$^{-1}$ were used. The sample was cooled at a rate of 0.5 K h$^{-1}$ while AFM scanning was carried on the *h*-BN bubbles. The highest temperature at which the bubbles became flat was recorded as the transition temperature. Afterwards, the samples were warmed up for several Kelvins to observe the expansion of bubbles in order to exclude the possible case of gas leakage.

**Scanning transmission electron microscopy**. Electron microscopy experiments were conducted using a Nion UltraSTEM100 scanning transmission electron microscope, operated at 60 kV in near-ultrahigh vacuum ($2 \times 10^{-7}$ Pa). The beam current during the experiments varied between 8 and 80 pA, corresponding to dose rates of ~5–50 × 10$^7$ e Å$^{-2}$ s$^{-1}$. The beam convergence semi-angle was 35 mrad and the semi-angular range of the medium-angle annular-dark-field detector was 60–200 mrad.

## Data availability
The data that support the findings of this study are available from the corresponding authors on request.

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

## Acknowledgements

The work was partially supported by the National Key R&D program (Grant No. 2017YFF0206106), the Strategic Priority Research Program of Chinese Academy of Sciences (Grant No. XDB30000000), the National Science Foundation of China (Grant Nos. 51772317, 51302096), the Science and Technology Commission of Shanghai Municipality (Grant No. 16ZR1442700), the Hubei Provincial Natural Science Foundation of China (Grant No. ZRMS2017000370), and the Fundamental Research Funds of Wuhan City (No. 2016060101010075). K.W. and T.T. acknowledge support from the Elemental Strategy Initiative conducted by the MEXT, Japan and JSPS KAKENHI Grant Numbers JP15K21722. C.L. acknowledges support from the European Union's Horizon 2020 research and innovation program under the Marie Skłodowska-Curie Grants No. 656378—Interfacial Reactions. L.H. acknowledges financial support from the program of China Scholarships Council (No. 201706160037). H.W. and D.Z. thank Y. Gu, Y. Ma, X. Chen (Shanghai Institute of Technical Physics, Chinese Academy of Sciences) for FTIR spectra measurement. L.C. and L.H. thank Q. Liu and Z. Liu (Shanghai Institute of Microsystem and Information Technology, Chinese Academy of Sciences) for measurement in XPS spectra and mass spectra.

## Author contributions

H.W. conceived and designed the research. L.H. fabricated the samples and performed the measurements. L.C., C.Z. and Huishan W. assisted with the transfer technique and AFM measurements. C.J. carried out low-temperature AFM measurements. X.W. and H.X. provided assistance with substrate fabrication. Z.W., W.W. and Z.N. performed the Raman measurements. C.L., K.E., J.M. carried out STEM measurements. K.W. and T.T. fabricated the *h*-BN crystals. L.H., C.L. X.X., X.M., D.Z. and Haomin W. prepared the manuscript. All authors discussed the results and commented on the manuscript.

## Additional information

**Competing interests:** The authors declare no competing interests.

