## [Peer Review File · Nature Communications]

Response Letter

Round-1 Feedback:

Reviewers' comments:

Reviewer #1 (Remarks to the Author):

The paper by He et al. described the Isolating of hydrogen in hexagonal boron nitride bubbles by a plasma treatment. This work is a good contribution to the field and could be published in Nature Communication after major revision as mentioned below:

1. The paper contains some topographical errors: please correct
2. In order to prove their theory on the confinement of H₂ by the hBN, several experiments should be performed and added to the paper:
 - a. The authors did not give any chemical or structural characterizations of the hBN layers. The authors should show the influence of plasma treatment in the composition of hBN. To prove that the plasma treatment under different gazes and different conditions does not impact the composition of hBN and does not induce porosity in the hBN layers, at least XPS, TEM/EELS, FTIR, STM etc. should be performed and added to the manuscript.
 - b. A depth profile XPS or cross section TEM/EELS should be as well performed and added to the SI to prove that Atomic hydrogen does not impact the composition of the quartz support.
 - c. Mass spectrometry should be as well performed on the mixture of gas before and after extraction of hydrogen to prove that this technique can extract hydrogen from other gases in an efficient way.

Reviewer #2 (Remarks to the Author):

The article “Isolating hydrogen in hexagonal boron nitride bubbles by a plasma treatment” by Li He et al addresses an interesting topic, i.e., formation of hydrogen filled bubbles with controllable radius on hBN surface. Using RF hydrogen plasma as a tool, the authors demonstrated the feasibility of hydrogen extraction from hydrocarbons and mixed gases. Material analysis of the treated samples that includes, optical microscopy, atomic force microscopy and Raman spectroscopy has only been carried out. The article presents interesting results about extracting hydrogen from gaseous mixtures applying plasma treatment. However, some drawbacks need to be pointed out, namely:

- i) The characterization of the main tool, i.e. RF hydrogen plasma, used to extract hydrogen and to form bubbles on hBN is entirely missing. The authors should justify their choice to use plasma.
- ii) There are no references about plasma treatment of 2D materials in the introduction part. Plasma treatment is widely used for processing of materials, including 2Dmaterials.
- iii) In page 4, RF generator should replace "microwave generator".
- iv) "Only accelerated hydrogen atoms possess sufficient kinetic energy to form bubbles" claim the authors. However, there is no estimation about the energy of H atoms involved in the bubbles

formation. The obtained results suggest that the energy and number density of energetic H atoms are crucial parameters for the process. To this end, information about electron temperature and degree of hydrogen dissociation in low-pressure RF plasma is essential.

v) The dc electric fields in the sheath domain near the walls of the plasma reactor and around the sample under treatment can play a major role in the process. Energetic H atoms can originate from charge (H⁺ ions) acceleration followed by wall neutralization. At lower pressures, the ion mean free paths can be larger than the radius of plasma reactor and H⁺ ions acquire a kinetic energy equal to the radial potential energy drop, which is of the order of electron temperature (see for instance Felizardo et al Appl. Phys. Lett. 99, 041503 (2011); doi:10.1063/1.3621824). This issue should be clearly addressed.

vi) Information about plasma reactor dimensions (e.g. radius) and dimensions of the treated sample and spatial orientation should be provided. How the sample was supported in the plasma volume?

i) Degree of dissociation, i.e. density of H atoms, is essential to explain the results about extraction of hydrogen from gas mixtures and consequent isolation between hBN layers in bubbles. Moreover, formation of the pits on hBN surface depends on the number density and kinetic energy of H atoms impinging the surface. This issue should be addressed.

ii) The authors claim, "Normally, the density and size of bubbles depend on the condition of the plasma treatment". Therefore, detail analysis about plasma driven kinetic mechanisms behind the process considered should be provided. The main plasma based mechanisms should be pointed out.

The article needs major revision along the above-mentioned guidelines.

Reviewer #3 (Remarks to the Author):

In this manuscript, He et al. exposed hexagonal boron nitride (hBN) crystals exfoliated onto quartz substrates to mild plasmas of oxygen, argon and hydrogen. When hydrogen plasmas are used, they observe the formation of blisters in the crystals. But for plasmas of oxygen or argon they do not. Increasing the time the crystals were exposed to the hydrogen plasma, reduced the area density of the blisters, but yielded larger blisters. A similar effect was found when the temperature of the substrate was increased for a fixed exposure time.

The manuscript attributes these observations to the formation of hydrogen molecules inside the crystals, which expand the interlayer crystal space to form the observed blisters. While this seems a reasonable conclusion, the manuscript does not discuss the possibility of other gases being formed by the reaction of the hydrogen plasma with the crystal. Some extra effort to prove that the trapped gas is indeed hydrogen would improve the manuscript. For example, a recent arXiv paper (Tedeschi, et al. arXiv:1803.09825) reached similar conclusions on TMDC's. And to prove that the blisters indeed contain hydrogen, they showed that the bubbles suddenly collapse at 30 K (within 10mK).

Next, the manuscript reports the stability over time of the blisters. The diameter of the bubbles decreased slightly as a function of time. This was attributed to hydrogen leaking out of the bubbles. Again, there is no dedicated effort to prove this claim. For example, Fig. 3 should have

included the height of the bubbles as well as the diameter. The important parameter is the blister volume, not the diameter. Presumably, the claim is that the aspect ratio of the bubbles is roughly constant – as plotted in Figure 2d – and hence the volume is decreasing when the diameter decreases. This unfortunately is not convincing, given that the error in determining the aspect ratio of the blisters (right axis Fig. 2d) is large. The possibility of other phenomena behind the observation is not discussed. For example, do the bubbles just expand and the drop in height is below the experimental resolution? This is not inconceivable given that the manuscript demonstrates that bubbles can merge with each other. In contrast, Tedeschi et al. demonstrated no leakage at all for over a year. I suspect that it is unlikely that leakage of hydrogen out of the bubbles is responsible for the observation of decrease in bubble diameter over time.

Some minor suggestions are the following. First, the abstract claims “an effective method for extracting hydrogen from gaseous compounds...”. This is perhaps misleading. If all the assumptions of the paper are accepted, the manuscript demonstrates a method to store hydrogen. The extraction itself is made by the plasma. Second, the schematics in Figs. 1 and 6 show blisters containing gases trapped by hBN and the quartz substrate. This is misleading. The gases are completely contained by hBN, otherwise they would leak through the quartz substrate (Buch et al, Nano Letters 2008). Third, references 12 and 14 are repeated.

With these comments in mind, I cannot recommend the publication of the manuscript in its present form. Further work is needed to provide a convincing interpretation of the data. This is a shame because some of the observations in the manuscript would be of interest to a wide audience.

Point-to-point Response to Round-1 Feedback:

Reviewer #1

General Comment: *The paper by He et al. described the Isolating of hydrogen in hexagonal boron nitride bubbles by a plasma treatment. This work is a good contribution to the field and could be published in Nature Communication after major revision as mentioned below.*

Reply: We appreciate the referee's positive comments on our research direction and experimental attempt. We carefully went through all the comments and performed more detailed investigations. We hope that the revised version is more convincing. Pls find our point-by-point reply to your concerns.

Comment 1: *The paper contains some topographical errors: please correct.*

Reply: We have carefully checked our manuscript and revised the topographical errors accordingly.

Comment 2: *In order to prove their theory on the confinement of H₂ by the h-BN, several experiments should be performed and added to the paper:*

a. *The authors did not give any chemical or structural characterizations of the h-BN layers. The authors should show the influence of plasma treatment in the composition of h-BN. To prove that the plasma treatment under different gazes and different conditions does not impact the composition of h-BN and does not induce porosity in the h-BN layers, at least XPS, TEM/EELS, FTIR, STM etc. should be performed and added to the manuscript.*

Reply: We would like to thank the reviewer for his/her valuable comments. As there are several concerns/suggestions raised, we will response them accordingly. By following the referee's suggestion, additional experiments including XPS, TEM, EDX and FTIR were carried out to characterize the structure and composition of h-BN bubble before and after plasma treatment. Detailed experimental results are presented below. BTW, as available TEM systems are only equipped with a EDX detector. Only the EDX data were used to analyze the composition of h-BN, instead of a EELS detector. In addition, h-BN is an insulator which is not suitable for STM characterization, STM measurement was not conducted on h-BN.

1) X-ray photoelectron spectroscopy (XPS)

Firstly, X-ray photoelectron spectroscopy (XPS) measurement was carried out to investigate the influence of H₂ plasma treatment on h-BN flakes. Substrates with h-BN flakes are measured on SPECS XPS system using monochromatic Mg K α line at a base pressure of 10⁻⁹ mbar before and after plasma treatment. The XPS survey analysis in the full energy range is shown in Fig. R1. XPS spectra of h-BN flakes are calibrated with reference to C 1s at 284.5 eV. The specific measurement results are given in Fig. R2. Fig. R2a shows the narrow scan B 1s of h-BN before and after plasma treatment. As shown in Fig. R2a, there is an obvious peak in the binding energy range from 184 eV to 196 eV, which can be fitted into two peaks. The main peak (the green dashed curve) corresponds to B-N bond of B 1s, while the tiny peak (the red dashed curve) corresponds to the B-O bond which were caused by annealing in O₂ flow at 600 °C. Annealing in

O₂ flow is necessary to remove tape residues after exfoliation. Fig. R2b shows the N 1s core level spectra that were fitted to only one curve which corresponds to B-N bond. It is clear that the XPS spectra of *h*-BN flakes do not change obviously even after H₂ plasma treatment. As shown in Table R1, the *h*-BN XPS peaks with bubbles have broadened, this phenomenon can be reasonably attributed to the *h*-BN expansion caused by bubbles formation.

Figure R1 | XPS survey analysis of *h*-BN flakes on quartz substrate before (a) and after (b) H₂ plasma treatment in the full energy range. The B1s and N1s peaks are visible. Inset shows a quartz substrate with *h*-BN flakes placed on a sample holder.

Figure R2 | Fine XPS spectra survey of *h*-BN flakes before and after H₂ plasma treatment. B 1s (a) and N 1s (b) spectra obtained from *h*-BNs flakes. Dashed lines are fitting curves.

	B-N(FWHM) eV	B-O(FWHM) eV	N-1s eV
h -BN with bubbles (after plasma treatment)	190.4(1.24)	191.5(0.8)	398.1(1.44)
h -BN (before plasma treatment)	190.3(1.17)	191.3(0.8)	398.1(1.29)

Table R1 | Peak position and FWHM of XPS spectra of the *h*-BN flakes before and after plasma treatment.

2) Transmission Electron Microscopy (TEM) measurement

A group of cross sectional images measured by scanning electron microscope (SEM) and TEM are presented to characterize the *h*-BN bubble structure. The specimen for TEM measurement was prepared by Focused Ion Beam (FIB) system (Helios NanoLab 600) which is equipped with a function of SEM. Fig. R3 shows the specific processes to make a specimen of *h*-BN bubble. The green rectangle indicated in Fig. R3c is the area selected for e-beam induced Pt deposition. After Pt deposition (Fig. R3e), the TEM specimen for bubble cross-sectional imaging was ready after FIB shaping (Fig. R3f, R3g, Fig.R4a). The SEM images are captured by the SEM in the FIB system. Then, the specimen was transferred to a TEM chamber (JEOL 2100F, operated at 200 kV). The enlarged cross-sectional scanning images by TEM are shown in a set of pictures which are manually combined to exhibit the *h*-BN bubble from the middle to the border area (Fig. R3b-f). Noted that the gap of the bubble is so narrow (in a few tens of nm) that we can hardly see it in the overall cross-sectional TEM image of Fig. R3a.

Figure R3 | Optical Microscopy (OM) and SEM images acquired during TEM specimen preparation. (a-b) Optical images of *h*-BN flakes with bubbles. The sequence of SEM images (c to h) corresponds to the chronological steps required for the fabrication of a *h*-BN bubble TEM specimen. The specimen was prepared in a dual-beam system (Helios NanoLab 600). (c-e) the SEM images of the top view of the *h*-BN flake with bubbles. (f-h) Cross-sectional SEM image of the specimen after FIB milling.

Figure R4 | Enlarged cross-sectional TEM view of bubble position inside *h*-BN. (a) an overall cross-sectional picture of the *h*-BN bubble-specimen after FIB fabrication; (b-f) Specific enlarged TEM images from middle to border area of *h*-BN bubble-position.

3) Energy Dispersive X-Ray (EDX) analysis on elemental composition of *h*-BN

EDX is a semi-quantitative x-ray technique used to identify the elemental composition of materials. The scanning transmission electron microscope (STEM) is equipped with a EDX

detector. Fig. R5 shows EDX elemental analysis of the cross-sectional interface of *h*-BN specimen. As shown in Fig. R5b, EDX spectrum shows peaks corresponding to the elements (B, N, C and Pt), where B and N the main elements. Elemental mapping of a sample and image analysis are given in Fig. R5c-e. EDX elemental mapping clearly shows that the bubble area is empty.

Figure R5 | EDX elemental analysis of the cross-sectional interface of *h*-BN specimen. (a) Scanning transmission electron microscope (STEM) image shows the selected EDX inspection area. *h*-BN and Pt area are indicated. **(b)** The resulting spectrum and tabulated results reveal that B, N, and Pt are the main elements present with C element being mainly caused by hydrocarbon contamination. **(c-e)** The corresponding elemental mapping of N, B and Pt, respectively. EDX elemental mapping indicates that the bubble area is empty.

4) Fourier-transform infrared (FTIR) spectra

Hydrogen plasma treatment was carried out on *h*-BN flakes exfoliated on pristine <100> silicon substrate to produce bubbles. FTIR measurement was conducted at ~1 mbar (on Bruker IFS 66v/S) before and after hydrogen plasma treatment. The FTIR spectra of the *h*-BN before and after H₂ plasma treatment are shown in Fig. R6. There are 3 obvious dips in the range from 500 to 4000 cm⁻¹. The dip at 1369 cm⁻¹ was derived from the in-plane stretching vibration of *h*-BN, and the dip at 817cm⁻¹ was owing to out-of-plane bending absorption of the B-N-B. The absorption peak appears at 1045 cm⁻¹ was a fingerprint of the presence of Si-O bonds, which was derived from the substrate. The similar characteristics in FTIR spectra indicate that the plasma treatment did not chemically cause obvious change in *h*-BN flakes.

Figure R6 | FTIR spectra of the *h*-BN flakes on silicon substrates before and after H₂ plasma treatment.

Comment 3: b. A depth profile XPS or cross section TEM/EELS should be as well performed and added to the SI to prove that Atomic hydrogen does not impact the composition of the quartz support.

Reply: We would like to thank the reviewer for his/her valuable comments. We guess that the reviewer may want us to exclude possible cases that the bubbles were caused by the deformation or components' variation of the quartz substrate. Experimental results given in Fig. R3g-h and Fig. R4 show that the interface between *h*-BN and quartz substrate are flat. We believe that the bubble formation has nothing to do with the quartz substrate. Besides, the cross-sectional images shown in Fig. R4-R5 indicate that the bubbles locate at the interlayers of the *h*-BN instead of the interface between *h*-BN and the quartz substrate. Element line profile scans for Pt, N and B using EDX on STEM were performed at the STEM specimen of *h*-BN bubble. A depth profiles of EDX confirm that the bubble area is in-between *h*-BN layers.

Figure R7 | Element line profile scan for Pt, N and B using EDX on STEM. (a) High-resolution STEM cross-sectional image of a *h*-BN bubble. **(b)** The lateral distribution of Pt, N and B by using EDX analysis via line scan.

Comment 4: *c. Mass spectrometry should be as well performed on the mixture of gas before and after extraction of hydrogen to prove that this technique can extract hydrogen from other gases in an efficient way.*

Reply: We would like to thank the reviewer for his/her valuable comments. To measure the mass spectra before and after extraction of hydrogen is of course interesting. However, the residue gas analysis (RGA) is very difficult to perform in the existing experimental setup, because of the limitation of the amount of *h*-BN flakes. The experiment may need a large number of *h*-BN flakes, e.g. several grams, and very complex design of their supporting. We only have several tiny *h*-BN flakes in size of < 1mm. The proportion of the extracted hydrogen, compared to the continuous hydrogen flow in the tube, will be very low.

Actually, mass spectrometry measurement can be conducted to analyze gases in the bubble. Mass spectra (MS) analysis was carried out on a UHV vacuum system with quadrupole mass spectrometer (Pfeiffer QMS 220 M). The sample for MS measurement is a quartz substrate in size of 10 × 10 mm with a high density of *h*-BN flakes, which has very dense bubbles (Fig. R8). The measurement system is shown in Fig. R9a and the schematic of its structure is given in Fig. R9b. The substrate with *h*-BN bubbles were installed in a homemade sample holder (see Fig. R9c). As shown in Fig. R9c, another silicon substrate with rough surface were placed on the quartz substrate with *h*-BN bubbles surface, and both substrates were assembled on sample holder gently. The holder has a screw which can slide the top silicon substrate. The sliding can spoil the *h*-BN bubbles on underlying quartz substrate. The experiment was carried out in a UHV chamber with a base pressure of 10⁻⁹ mbar. After the base pressure in the chamber became stable, we rotated the screw to break *h*-BN bubbles. The mass spectrum variation was recorded and plotted in Fig. R9d. We set several possible gases (H₂, Ar, NO₂, N₂ and NH₃) for monitoring. As shown in Fig. R9d, only the hydrogen increased suddenly after the bubbles were broken, and then gradually decreases with time. This experiment proves that the gas inside the bubble is indeed hydrogen.

Here, we discuss some details about the experimental design. The density of *h*-BN flakes on 10×10 mm quartz substrate is about 30×30 pieces, and the density of bubbles on each piece of *h*-BN flake is about 100×100. Assuming the pressure in a bubble is about 10 bar, and the volume of the bubble is about 10 μm × 10 μm × 0.1 μm. The volume of the chamber is 0.1 m³. Even if 1% of bubbles were broken, the pressure will reach about 9 × 10⁻⁸ mbar. Compared with the base pressure of chamber, 10⁻⁹ mbar, the gas is sufficient to be detected.

Figure R8 | Optical microscope image of a *h*-BN flake with dense bubbles on a quartz substrate, which is the sample prepared for the mass spectra measurement.

Figure R9 | Experimental details about the mass spectra analysis on *h*-BN bubbles. (a) A UHV system equipped with a mass spectrometer. (b) Schematic for the vacuum system. (c) sample holder used for the mass spectrum analysis. (d) Real-time monitoring on different gases by mass spectrometer.

Reviewer #2:

General Comment: *The article “Isolating hydrogen in hexagonal boron nitride bubbles by a plasma treatment” by Li He et al addresses an interesting topic, i.e., formation of hydrogen filled bubbles with controllable radius on h-BN surface. Using RF hydrogen plasma as a tool, the authors demonstrated the feasibility of hydrogen extraction from hydrocarbons and mixed gases. Material analysis of the treated samples that includes, optical microscopy, atomic force microscopy and Raman spectroscopy has only been carried out. The article presents interesting results about extracting hydrogen from gaseous mixtures applying plasma treatment. However, some drawbacks need to be pointed out, namely.*

Reply: We greatly appreciate the referee’s positive evaluation of our work. The valuable comments helped us improve our manuscript in many ways.

Comment 1: *The characterization of the main tool, i.e. RF hydrogen plasma, used to extract hydrogen and to form bubbles on h-BN is entirely missing. The authors should justify their choice to use plasma.*

Reply: We would like to thank the referee for this comment. The specific characterization of the main tool is given in Fig. R10. Plasma state is generally known as one of the four fundamental states of matter, in which the neutral gases could be converted to ionized gases. Here, we use a cylindrical inductively coupled plasma (ICP) system to generate plasma state of hydrogen gases, in which there are plenty of protons that could possibly penetrate the 2D mesh of h-BN, and finally recombined to hydrogen molecules in the interlayers of h-BN. Actually, ion implantation equipment can do similar thing in more precise way, but the cost of experiment will be relatively high.

Figure R10 | Illustration of the experimental setup for the fabrication of hydrogen bubbles on *h*-BN. (a) Front view of the plasma chamber system for bubble fabrication, where L_1 (110 cm) is the length of the plasma tube, L_2 (60 cm) is the distance from the entrance to the *h*-BN sample, L_3 (30 cm) is the horizontal distance between the RF coil and the *h*-BN sample and L_4 (20 cm) represents the length of the RF coil. (b) Lateral photograph of the Plasma system corresponding to (a), captions in the image shows different functional units of the plasma system. (c) Cross-sectional view of the plasma chamber system for bubble fabrication, where d_1 (12 cm) denotes the diameter of the RF coil, d_2 (5 cm) is the outer-wall diameter of the plasma tube, d_3 (4.2 cm) is the inner-wall diameter of the plasma tube, T_1 (0.8 cm) is the thickness of the wall of the plasma tube, T_2 (0.7 cm) is the thickness of the corundum substrate for sample (the *h*-BN sample is exfoliated onto a quartz substrate which is placed on the corundum substrate), w_1 (3.1 cm) is the width of the corundum substrate and finally h_1 (1.9 cm) represents the height of the

sample from the bottom of the tube. **(d)** Cross-sectional photograph of the entrance of plasma tube and the corundum sample holder.

Comment 2: *There are no references about plasma treatment of 2D materials in the introduction part. Plasma treatment is widely used for processing of materials, including 2D materials.*

Reply: We would like to thank the referee for the suggestion. Recent progresses about plasma treatment on 2D materials are included (the plasma treatment includes etching, growth and hydrogenation) into the introduction of our manuscript.

Comment 3: *In page 4, RF generator should replace "microwave generator".*

Response: We would like to thank the referee for the suggestion. By searching through internet, we understood the difference between “RF” and “microwave”. The replacement has been done accordingly.

Comment 4: *"Only accelerated hydrogen atoms possess sufficient kinetic energy to form bubbles" claim the authors. However, there is no estimation about the energy of H atoms involved in the bubbles formation. The obtained results suggest that the energy and number density of energetic H atoms are crucial parameters for the process. To this end, information about electron temperature and degree of hydrogen dissociation in low-pressure RF plasma is essential.*

Reply: We would like to thank the referee for this valuable comment. To understand the crucial parameters for the processes e.g. the energy and number density of energetic H atoms, electron temperature and degree of hydrogen dissociation, we carried out the simulation of the plasma in tube by Comsol Multiphysics. In this software, we try to simulate distributions of different parameters in the reaction tube, whose kinetic processes are based on non-equilibrium hydrogen plasma (Matveyev AA et al. Kinetic processes in a highly-ionized non-equilibrium hydrogen plasma. Plasma Sources Science and Technology. 4, 606-617 (1995)). The bottom of the tube in the simulation images represents the gas entrance of the tube chamber while the gas flows upward. The sample position has been marked on Fig. R11b, d, e (~0.6 m on the z-axis), where the temperature is about 350 °C around. The simulation study has been performed with a typical hydrogen flow rate fixed at 3 sccm, back-ground pressure kept at 3 Pa, a RF power of 100W and an inductor frequency of 13.56 MHz. All these parameters are the same with those used in our experiments. The simulations generally depict the distribution of plasma species within the reaction tube. The **electron density** (Fig. R11a) attains its maximum value ($\sim 5.5 \times 10^{17} \text{ m}^{-3}$) in the plasma core region, and drops near the wall, meanwhile, this value drops to $\sim 2.5 \times 10^8 \text{ m}^{-3}$ at the sample position (Fig. R11b). In the most intense regions, inside the plasma tube, the electron density evolution is similar to the **ion density evolution** ($\sim 5.5 \times 10^{17} \text{ m}^{-3}$) (Fig. R11a, 11c). But at the region that is far from the plasma core, due to the steep electron gradients, the ion density ($\sim 0.9 \times 10^{10} \text{ m}^{-3}$) becomes greater than the electron density ($\sim 2.5 \times 10^8 \text{ m}^{-3}$) (Fig. R11d). Fig. R11e shows the calculated **electrons temperature** in the plasma reactor. This Figure reveals that the

maximum temperature of electrons is approximately equal (about ~ 3.1 eV) in the heart of the plasma region but dropping down to ~ 0.6 eV at the sample position. The energy of H ions is estimated based on the electron temperature, which will be presented in the reply to Comment 5 and the degree of hydrogen dissociation will be given in the reply to Comment 7.

Figure R11 | Distribution of (a-b) electron density, (c-d) ion density and (e) electron temperature in hydrogen plasma reaction tube, simulated via Comsol Multiphysics.

Comment 5: *The dc electric fields in the sheath domain near the walls of the plasma reactor and around the sample under treatment can play a major role in the process. Energetic H atoms can originate from charge (H^+ ions) acceleration followed by wall neutralization. At lower pressures, the ion mean free paths can be larger than the radius of plasma reactor and H^+ ions acquire a kinetic energy equal to the radial potential energy drop, which is of the order of electron temperature (see for instance Felizardo et al Appl. Phys. Lett. 99, 041503 (2011); doi:10.1063/1.3621824). This issue should be clearly addressed.*

Reply: We would like to thank the referee for this valuable comment.

We are very sorry that our experimental setup was missing in our previous manuscript. It looks the referee thought that our setup includes an ion source and a dc electric field accelerator. Actually, our experiment setup is just a simple tube furnace equipped with a ICP generator. However, it does not affect our trial in estimation on kinetic energy of H^+ ions.

Our PECVD furnace does not support installation of an additional flange for an inner detector to measure electron temperature in real time. Fortunately, we found a similar equipment installed with a probe for electron temperature measurement. In the “new” equipment, the process parameters are adjusted to the similar values in our “old” PECVD furnace to produce hydrogen bubbles on *h*-BN via plasma treatment. After that, the electron temperature was measured via a Hiden ESPion Langmuir probe placed at the plasma core region inside the reaction tube. The probe mounts to the chamber via a DN-35-CF UHV flange (2 3/4” conflat) when the H-plasma system is working. This test gives an average electron temperature $T_e = 3.28 \pm 0.5$ eV. Assuming that all these dissociated hydrogen ions are H^+ , the estimated average ionic mass of 1 a.m.u. (1/12 m of ^{12}C , $\sim 1.660539040 \times 10^{-27}$ kg) is adopted to calculate the initial energy of the ions (ϵ_i) as well as the energy for ions to go across the plasma sheath. According to the equation that ϵ_i

$= (T_e / 2) \ln (M / 2\pi m) + 0.5 (T_e)$. (Godyak VA et al. Electron energy distribution function measurements and plasma parameters in inductively coupled argon plasma. Plasma Sources Science and Technology. 11, 525-543 (2002)), where T_e represents the electron temperature and M (1.66×10^{-27} kg) and m (9.1×10^{-31} kg) are the ion and electron mass, we estimate that the ε_i is $\sim 3.34 T_e$ for H^+ , which is $\sim 10.96 \pm 1.7$ eV at the core plasma region and ~ 2.0 eV at the sample region.

Comment 6: *Information about plasma reactor dimensions (e.g. radius) and dimensions of the treated sample and spatial orientation should be provided. How the sample was supported in the plasma volume?*

Reply: Thank you for this kind suggestion. The schematic of experimental setup has been given in Fig. R10. All the experiments are carried in a normal tube furnace. As shown in Fig. R10d, the quartz substrates with *h*-BN flakes were placed on a corundum holder in the tube. The center of plasma generator is about 30 cm away from the *h*-BN sample in a corundum tube, where there is a set of inductive coils rolled around the cylindrical chamber. A 13.56 MHz RF generator with the matching network is connected to the coils with a typical working power of 100W. The substrates with *h*-BN flakes lied on a corundum holder, which is located at the center of the chamber where there is a heating furnace. Details about the plasma reactor have been added to supplementary information.

Comment 7: *Degree of dissociation, i.e. density of H atoms, is essential to explain the results about extraction of hydrogen from gas mixtures and consequent isolation between h-BN layers in bubbles. Moreover, formation of the pits on h-BN surface depends on the number density and kinetic energy of H atoms impinging the surface. This issue should be addressed.*

Reply: We would like to thank the reviewer for his/her valuable comments.

We try to estimate both the density of H ions and the degree of hydrogen dissociation in the hydrogen plasma experiment via simulation. As the shown in Fig. R11c-d, the H ions' density is about $0.9 \times 10^{10} \text{ m}^{-3}$ at the sample position and $\sim 5.5 \times 10^{17} \text{ m}^{-3}$ at the plasma core region. The pressure distribution in plasma reactor was simulated and presented in Fig. R11g. The initial pressure value we input for the simulation is 3.05 Pa, which is corresponding to the value our pressure gauge detected before we turn on the RF system. The simulation result shows that the pressure at the sample position is ~ 1.2 Pa after launching the plasma system. This value helps us to estimate the degree of hydrogen dissociation, which ranges from 2.5% to 12.5% in our H-plasma process, according to the reference specified in the degree of hydrogen dissociation in a low-pressure plasma source. (Iordanova S *et al.* Hydrogen Degree of Dissociation in a Low Pressure Tandem Plasma Source. Spectroscopy Letters. 44, 8-16 (2011).)

Comment 8: *The authors claim, "Normally, the density and size of bubbles depend on the condition of the plasma treatment". Therefore, detail analysis about plasma driven kinetic mechanisms behind the process considered should be provided. The main plasma based mechanisms should be pointed out.*

Reply: We would like to thank the reviewer for his/her valuable comments.

By following the suggestion, we try to do detail analysis about plasma driven kinetic mechanisms. Fig. R12 shows the collateral effects of some different physical parameters result from the variation of the heating temperature by Comsol simulations. The simulations exhibit some differences of spacial distribution of temperature in the plasma tube when the sample position was set to 30 °C and 350 °C (Fig. R12 (a-b)). It is obvious that when the temperature increases, the electron density is decreased along with the ion density (Fig. R12 (c-d), (e-f)), the electron temperature is slightly decreased but almost keep the same (Fig. R12 (g-h)), and the pressure and velocity at the sample position are both increased (Fig. R12 (i-j), (k-l)). From the simulation results, we found that the ion density decreases while the thermal dynamic velocity of ions increases. This could be a possible reason to explain why the density of bubble decreases and the size of bubbles increase when the samples were heated up (See in Fig. R16c in main manuscript).

Figure R12 |Distribution contrast of different physical parameters in plasma tube between room temperature (30 °C, 303 K) and 350 °C (623 K) at the sample position. (a-b) temperature, (c-d) electron density, (e-f) ion density, (g-h) electron temperature, (i-j) pressure, and (k-l) velocity.

Reviewer #3:

Comment One: *The manuscript attributes these observations to the formation of hydrogen molecules inside the crystals, which expand the interlayer crystal space to form the observed blisters. While this seems a reasonable conclusion, the manuscript does not discuss the possibility of other gases being formed by the reaction of the hydrogen plasma with the crystal. Some extra effort to prove that the trapped gas is indeed hydrogen would improve the manuscript. For example, a recent arXiv paper (Tedeschi, et al. arXiv:1803.09825) reached similar conclusions on TMDC's. And to prove that the blisters indeed contain hydrogen, they shew that the bubbles suddenly collapse at 30 K(within10mK).*

Reply: We would like to thank the reviewer for his/her valuable comments.

To verify the trapped gas is indeed hydrogen inside *h*-BN bubbles is very important. We appreciate the referee's reminder on recent progress on TMDC bubbles containing hydrogen (Tedeschi, et al. arXiv:1803.09825). Fortunately, a low-temperature scanning probe microscopy (SPM) system was setup in our lab at the beginning of this year. It allows us to measure the temperature dependence of topographic variation on *h*-BN bubbles.

During the cooling process from room temperature, we found that bubbles shrink gradually. It can be easily understood by the second law of thermodynamics, where pressure inside the bubble mainly depends on the elastic modulus of *h*-BN. It is known that elastic modulus always keeps constant even when the temperature varies. As a result, the volume of the bubble will decrease during the cooling process. Wrinkles on *h*-BN surfaces may form when the bubbles shrink.

The topographies of *h*-BN bubbles have been recorded during cooling, and the results were plotted in Fig. R13.

Figure R13 | Swelling and deflating process of the *h*-BN bubbles containing hydrogen. (a) An optical image of bubbles on a *h*-BN flake, the image was taken under ambient condition, scale bar: 5 μm; (b) Topographic AFM images of the red-box area shown in (a) measured at 34 K and 33 K respectively, the AFM images were taken in a vacuum chamber; (c) The height profiles of line-scan at the same place (indicated by dashed lines in (b)) where the bubbles remain at ~34 K and disappear at ~33 K. (d) Histogram of the transition temperature ($T_{\text{transition}}$) at which bubbles collapses. The red line is Gaussian fit to the data. The yellow line is the histogram cumulative function (right axis).

As shown in Fig. R13, $T_{\text{transition}}$ of all the bubbles measured can be described as Gaussian distribution, and exhibits an average value equal to 33.2 ± 3.9 K. This value is very close to $T_{\text{transition}}$ of H_2 (33.93 K) among all possible gases (see Table R1). The results strongly indicate that the gases inside the bubbles are hydrogen molecular.

	Boiling point	Critical point	
Hydrogen	20.271 K	32.938 K,	1.2858 MPa
Helium	4.222 K	5.1953 K,	0.22746 MPa
Nitrogen	77.355 K	126.192 K,	3.3958 MPa
Oxygen	90.188 K	154.581 K,	5.043 MPa
Argon	87.302 K	150.687 K,	4.863 MPa
Methane	111.65 K	190.6 K,	4.64 MPa

Table R1| The boiling points and liquid-vapor critical points of different gasses adapted from Wikipedia. [1]

References:

[1] Referred from wikipedia.org

Comment Two: *Next, the manuscript reports the stability over time of the blisters. The diameter of the bubbles decreased slightly as a function of time. This was attributed to hydrogen leaking out of the bubbles. Again, there is no dedicated effort to prove this claim. For example, Fig. 3 should have included the height of the bubbles as well as the diameter. The important parameter is the blister volume, not the diameter. Presumably, the claim is that the aspect ratio of the bubbles is roughly constant – as plotted in Fig. R16d – and hence the volume is decreasing when the diameter decreases. This unfortunately is not convincing, given that the error in determining the aspect ratio of the blisters (right axis Fig. R16d) is large. The possibility of other phenomena behind the observation is not discussed. For example, do the bubbles just expand and the drop in height is below the experimental resolution? This is not inconceivable given that the manuscript demonstrates that bubbles can merge with each other. In contrast, Tedeschi et al. demonstrated no leakage at all for over a year. I suspect that it is unlikely that leakage of hydrogen out of the bubbles is responsible for the observation of decrease in bubble diameter over time.*

Reply: We would like to thank the reviewer for these valuable comments.

In experiments designed for previous Fig. 3, the bubbles on *h*-BN (Type-851, which is a serial number to identify different batch of *h*-BN crystal) were measured with an optical microscope. We did not have any height information for the bubbles. Here, we fabricated new *h*-BN bubbles similar to those shown in former Fig.3, the topographies of three bubbles on the same *h*-BN flake were recorded by AFM once a week for 10 continuous weeks. It is found that two bubbles are almost unchanged in this 10-week period (Fig. R14a and 14b) while bubbles with a diameter of $\sim 1\mu\text{m}$ experience some shrinkage in this duration (Fig. R14c). The aspect ratio (h_{max}/R) of the

bubble does not change in this duration (Fig. R14d). It looks that the leakage of hydrogen out of the bubbles is related to local crystalline quality of *h*-BN.

Figure R14 | topographic variation of *h*-BN bubbles in different diameter. (a) dimensional variation of a *h*-BN bubble with an initial diameter of $\sim 3.4 \mu\text{m}$. **(b)** The size-variation of an *h*-BN bubble with an initial diameter of $\sim 2.2 \mu\text{m}$. **(c)** The size-variation of an *h*-BN bubble with an initial diameter of $\sim 0.9 \mu\text{m}$. **(d)** Statistics of the maximal height values and their corresponding aspect ratios (h_{max}/R) of the shrinking bubble shown in (c).

Fig. R15 shows that a bubble inside a ruptured outer bubble. The results indicate that hydrogen molecules in *h*-BN bubbles usually have in depth distribution rather than accumulate in a fixed depth. It is easily understood as atomic hydrogens in plasma also have a distribution in kinetic energy. The gas molecules accumulated at the superficial interlayers (van der Waals gaps) are more easily to escape via defects channels while those trapped in deep interlayers may have less chance to escape.

Figure R15 | AFM topographical image of the bubble-in-bubble structure.

We also tried the *h*-BN flakes made from another batch *h*-BN crystal (type-855 and type-856) to fabricate bubbles, no obvious change in topography was found in all bubbles in a 10-week period, even for the bubbles whose diameter are smaller than $1 \mu\text{m}$. In *h*-BN bubbles made from *h*-BN crystal (type-854), obvious change in topography was found in many bubbles in a 10-week period. Based on above results, the leakage of hydrogen out of the bubbles may be related to the

quality of *h*-BN crystal. The natural defects or the superficial defects created by plasma bombing may lead to the leakage of hydrogen trapped in the shallow interlayers. (Gas leakage via defects has been studied in graphene Nature Nanotechnology, 10, 785-790 (2015))

As the main finding is that the *h*-BN multilayer is transparent to atomic hydrogen in the manuscript, we decide to move the former Fig. 3 and corresponding description to supplementary information.

Minor Suggestion 1: *Some minor suggestions are the following. First, the abstract claims “an effective method for extracting hydrogen from gaseous compounds...”. This is perhaps misleading. If all the assumptions of the paper are accepted, the manuscript demonstrates a method to store hydrogen. The extraction itself is made by the plasma.*

Reply: We would like to thank the reviewer for his/her very careful reading and valuable suggestion. We agree with the referee that our statement “an effective method for extracting hydrogen from gaseous compounds...” could be misleading. Hydrogen storage is a very important research field which have developed for tens of years, and most research into hydrogen storage is focused on storing hydrogen as a lightweight, compact energy carrier for mobile applications. Our work just exhibits an interesting experiment about isolating hydrogen in hexagonal boron nitride bubbles by a plasma treatment and explores possible mechanisms behind. We replaced the sentence with “we successfully demonstrated extraction of hydrogen from gaseous compounds or mixtures containing hydrogen.”.

Minor Suggestion 2: *Second, the schematics in Figs. 1 and 6 show blisters containing gases trapped by *h*-BN and the quartz substrate. This is misleading. The gases are completely contained by *h*-BN, otherwise they would leak through the quartz substrate (Buch et al, Nano Letters 2008).*

Response: We would like to thank the reviewer for his/her very careful reading and valuable suggestion. We made revision accordingly.

Fig. 1 revised in article

Fig. 7 revised in article

Minor Suggestion 3: *Third, references 12 and 14 are repeated.*

Response: We would like to thank the reviewer for his/her very careful reading. The corresponding revisions are made. We double-checked the reference and ensure that all the citations are correct.

Final comment: *With these comments in mind, I cannot recommend the publication of the manuscript in its present form. Further work is needed to provide a convincing interpretation of the data. This is a shame because some of the observations in the manuscript would be of interest to a wide audience.*

Reply: We appreciate greatly the referee's positive evaluation of our results and encouragement. The valuable comments helped us in improving our manuscript. We hope that both the relies we made above and the revised manuscript provide more convincing interpretation to our observation. We will appreciate if you can reconsider our manuscript for publication.

Additional information:

Figure R16 | STEM image showing the edge-on view configuration of a *h*-BN multilayer. (a) Schematic of a STEM investigation of the *h*-BN multilayer, the incident electron beam was parallel with its $[11\bar{2}0]$ crystallographic axis (the zigzag edge of the *h*-BN layer). (b) Cross-sectional schematic of the armchair edge-on view of *h*-BN multilayer. The red atoms represent boron atoms while the blue ones represent nitrogen atoms. (c) STEM-MAADF image, post-processed by a Wiener Filter, showing the edge-on view configuration of a *h*-BN multilayer. (d) Profiles of image intensity along the red and blue arrows marked in (c), showing the stacking sequence in *h*-BN. Their average peak-to-peak distances are ~ 2.26 Å, which are in agreement with the lattice spacing of *h*-BN. The synchronism of the peak intensity along the arrows demonstrates the stacking structure of *h*-BN must be either AA' or AA.

The mechanism of unimpeded penetration of atomic hydrogen into the *h*-BN interlayers can also be ascribed to two additional reasons. The first one is the polarization of the electron density distribution in this kind of 2D material. Recent experiments have demonstrated that *h*-BN possesses the best “porosity” among 2D materials. The concentration of valence electrons around the N atoms results in strong polarization of the BN bonds, making *h*-BN much more porous than graphene and MoS₂. The second reason is the stacking order. It has been confirmed in the former work that bi- and tri-layered graphene are also non-conductive to protons due to their

staggered lattices (AB stacked), in which the electron cloud of the first layer clogs the pores of the second. In contrast, multilayered *h*-BN exhibit excellent proton conductivity probably due to its favorable of AA' stacking structure, which has been regarded as the most stable one among five possible structures (AA' /AA/AB/ AB' / A'B) in recent calculation. The aligned pores in the out-of-plane direction also allow protons to move unimpededly throughout the space. This characteristic structure of multilayer *h*-BN could allow application in hydrogen storage.

To study the stacking order of *h*-BN multilayer, scanning transmission electron microscopy (STEM) investigation was carried out. The *h*-BN flakes are firstly exfoliated onto a quartz substrate, and then annealed to remove the residue of resist in an oxygen flow at 800 °C. After cooling down to room temperature, a group of wrinkles are found on the surfaces. The wrinkles, which are found along the armchair direction of *h*-BN crystals, help us determine the cutting direction of focused ion beam (FIB). The specimen obtained for the STEM investigation is similar to that shown in Fig. R4a.

Fig. R16a illustrates that the incident electron beam (e-beam) for the STEM investigation is parallel with the $[11\bar{2}0]$ crystallographic axis of *h*-BN. The cross-sectional schematic of the armchair edge-on view of *h*-BN multilayer is presented in Fig. R16b. Fig. R16c is a Wiener filtered STEM medium angle annular dark-field (MAADF) image showing the edge-on view configuration of *h*-BN multilayer, which reveals the layer stacking of the *h*-BN. The intensity profiles along two selected lines (red/blue) in Fig. R16c are showed in Fig. R16d, exhibiting apparent synchronism of the peak positions. Besides, the average peak-to-peak distances measured in red and blue profiles are both ~ 2.26 Å, which are close to 2.166 Å expected in *h*-BN. The minor deviation of the spacing in STEM measurement might due to the limitation of the scale calibration of the instrument. This result shows that the stacking order of our *h*-BN is either AA' or AA in our *h*-BN multilayer, which is in line with earlier prediction.

Figure R17 | Plane-view STEM measurement for a *h*-BN multilayer. (a) Schematic of how the STEM electron beam is applied onto the *h*-BN sample. The electron beam penetrates the *h*-BN along its [0001] crystallographic plane. (b) Plane-view HAADF image of the multilayer *h*-BN sample. (c) Profile of intensity corresponding to the green arrow-area noted in (a), shows AA' stacking structure of the multilayer *h*-BN crystal. (if it were AA stacked, i.e. N above N and B above B, the two atomic columns would have a very different HAADF intensity; and any staggered layer arrangement would show up as intensity within the hexagons).

Despite the cross-sectional measurement by STEM, we present a plane-view HAADF image (shew in Fig. R17) of another *h*-BN multilayer, which also confirm *h*-BN multilayer is an AA' stacking structure. The investigation provides direct evidence that the multilayered planes in *h*-BN may not be staggered such that the electron cloud of each layer does not block the pores of the successive layer. As such, atomic hydrogen could tunnel through multilayered *h*-BN.

Round-2 Feedback:

Reviewers' comments:

Reviewer #1 (Remarks to the Author):

The authors addressed well all my comments; the paper could be published now in nature communications.

Reviewer #2 (Remarks to the Author):

The revised version of the articles uprisa a lot of questions.

The authors have used Comsol Multiphysics software and have presented results obtained by numerical simulations.

However, it not clear what type of particles transport in plasma is dominant and assumed for simulations, i.e. diffusion or free fall regime.

It is not clear the difference between Fig. 11a and Fig.11b. The electron density decreases 7 orders of magnitude close to the sample.

It is confusing to see 7 orders of magnitude difference in electron density obtained just by slightly heating at 30oC of substrate placed inside the plasmas (see Fig. 11a and 12 c). What is the reason for such enormous difference? Moreover, the authors claim 2 order of magnitude difference in electron and ion density in the sample region. Plasmas fundamental property is their quasineutrality, i.e. nearly equal electron and ion density. The quasineutrality is destroyed only in the shield regions arising, for example, near the tube walls. To this end, what is the role of distributed (along the plasma axis) dc electric potential in formation of accelerated ions? How large is the Debye length as a measure for deviation from quasineutrality in the configuration considered?

“It looks the referee thought that our setup includes an ion source and a dc electric field accelerator. Actually, our experiment setup is just a simple tube furnace equipped with a ICP generator. However, it does not affect our trial in estimation on kinetic energy of H⁺ ions. “
The reviewer is aware that this is not an ion source and have experience with ICP plasmas.

The sentence below is unclear:

“Both the duration and temperature of plasma treatment have prominent effects on the density and sizes of bubbles, which provide a feasible way to fabricate h-BN bubbles with controllable dimensions.” What means “temperature of plasma treatment” ?

The main mechanisms behind the bubble formation are not clarified in the article. This is obviously plasma driven effect. The heating applied just results in some redistribution and size changing. The authors should focus on plasma mechanisms in order to give reliable explanation of the phenomenon considered. For example, changing the position of the sample along the plasma afterglow is one possible shoot to illuminate the phenomenon and to reveal the role of accelerated ions.

Moreover, concerning hydrogen extraction from hydrocarbons, the explanations such as: “The difference in the dose of atomic hydrogen between the ethyne and methane plasmas could be a key factor that accounts for this discrepancy in the bubble formation phenomenon.” are unacceptable.

The article is unbalanced by focusing mainly on material characterization and leaving unexplained and hidden the main plasma mechanisms resulting in bubbles formation.

Keeping in mind the comments above, I cannot recommend the publication of the manuscript in its present form.

Reviewer #3 (Remarks to the Author):

The authors have thoroughly addressed all the reviewers' comments. I have no further comments, other than to congratulate the authors on their work and to recommend the manuscript for publication.

Point-to-point Response to Round-2 Feedback:

General Comment of Reviewer #2:

The revised version of the articles uprises a lot of questions.

Reply: We thank the reviewer for his/her careful examination on our work and for bringing very insightful questions about the main mechanisms behind bubble formation. We may not have perfect answers to some of them, but we tried our best to give the most logical explanations.

We also welcome the reviewer's criticism regarding the unbalance of our manuscript on discussion of mechanism, which has helped us to re-evaluate carefully the context of our manuscript and to improve its presentation in the revised manuscript.

Comment 1:

The authors have used Comsol Multiphysics software and have presented results obtained by numerical simulations. However, it not clear what type of particles transport in plasma is dominant and assumed for simulations, i.e. diffusion or free fall regime.

Reply: We would like to thank the reviewer for his/her valuable comments.

Our sample is placed ~30 cm away from the radio-frequency coil (as shown in Fig. RR-1). In the absence of magnetic field, the transport type by which charged species move to the walls depends on their mean free path λ as compared to the reactor's characteristic length L . When $\lambda > L$, the plasma is in the free fall regime. When $\lambda < L$, loss of charged species mainly occurs through diffusion to the walls. As all our experiments are conducted at a low pressure, we can estimate the magnitude of the mean free path of particle collision by the equation of $\lambda = kT/p\sigma$, where k is the Boltzmann Constant (1.38×10^{-23} Joule/K), T is the environmental temperature at the sample position (350 °C), p is the pressure (~3 Pa) and σ is the effective cross-section target area in the collision process, which is related to the collision background particles as well as the velocity. Although σ is pretty difficult to confirm in the complex process, we try to estimate the minimum value of λ by supposing the collision background particles to be the hydrogen molecules, which have a homotactic radius of 144.5 pm, and the proton has a negligible radius compared to that of hydrogen molecule. We estimate $\lambda_{minimum} = kT/p\sigma \approx 0.0437$ m, which is roughly equal to the diameter of our reaction tube (0.042 m, d_3 shown in Fig. RR-1c).

Thus, we believe that it should be a free fall process at the sample position when we conduct the experiment with the sample's environmental temperature of 350 °C.

Figure RR-1 | Illustration of the experimental setup for the fabrication of hydrogen bubbles on *h*-BN. **a**, Front view of the plasma chamber system for bubble fabrication, where L_1 (110 cm) is the length of the plasma tube, L_2 (60 cm) is the distance from the entrance to the *h*-BN sample, L_3 (30 cm) is the horizontal distance between the RF coil and the *h*-BN sample and L_4 (20 cm) represents the length of the RF coil. **b**, Lateral photograph of the Plasma system corresponding to (a), captions in the image shows different functional units of the plasma system. **c**, Cross-sectional view of the plasma chamber system for bubble fabrication, where d_1 (12 cm) denotes the diameter of the RF coil, d_2 (5 cm) is the outer-wall diameter of the plasma tube, d_3 (4.2 cm) is the inner-wall diameter of the plasma tube, T_1 (0.8 cm) is the thickness of the wall of the plasma tube, T_2 (0.7 cm) is the thickness of the corundum substrate for sample (the *h*-BN sample is exfoliated onto a quartz substrate which is placed on the corundum substrate), w_1 (3.1 cm) is the width of the corundum substrate and finally h_1 (1.9 cm) represents the height of the sample from the bottom of the tube. **d**, Cross-sectional photograph of the entrance of plasma tube and the corundum sample holder.

Comment 2:

It is not clear the difference between Fig. 11a and Fig.11b. The electron density decreases 7 orders of magnitude close to the sample. It is confusing to see 7 orders of magnitude difference in electron density obtained just by slightly heating at 30°C of substrate placed inside the plasmas (see Fig. 11a and 12 c). What is the reason for such enormous difference?

Reply: We would like to thank the reviewer for his/her valuable comments.

Actually, Fig. 11a is the same as Fig.11b. In order to see the details of electron density at plasma core area (Fig. 11a) and sample area (Fig. 11b) more clearly, we plotted them in separated figures by modifying the scale of electron density.

As such, if one wants to see the difference between Fig. 11a and 12 c, it is more clear to compare Fig.11b with Fig. 12c. We put the separated three figures (Fig. 11a, 11b and 12c) here together (Fig. RR2a–2c)). Temperature at sample position is 350 °C in both Fig. RR-2a and 2b, while it is 30 °C in Fig.RR-2c. It is clear that the electron density is just slightly decreased at the sample position when the temperature increases from 30 °C to 350 °C.

We revised the figure captions accordingly for clarity.

Figure RR-2 | Distribution of electron density at a, Plasma core area, and b, Sample area when the environmental temperature at the sample position is kept at 350 °C. c, Distribution of electron density at sample area when the environmental temperature at the sample position is 30 °C.

Comment 3:

Moreover, the authors claim 2 order of magnitude difference in electron and ion density in the sample region. Plasmas fundamental property is their quasineutrality, i.e. nearly equal electron and ion density. The quasineutrality is destroyed only in the shield regions arising, for example, near the tube walls. To this end, what is the role of distributed (along the plasma axis) dc electric potential in formation of accelerated ions? How large is the Debye length as a measure for deviation from quasineutrality in the configuration considered?

Reply: We would like to thank the reviewer for his/her valuable comments.

According to the experimental set-up shown in Fig. RR1a-d, the sample is placed 30 cm away from the RF coil (the core region of the plasma). Along this axial direction, there is a steep electron gradient which leads to the unbalance between the electron density and ion density, and thus, the quasi-neutrality of plasma is not sustained. (see in Fig. RR-3a, b)

As the reviewer #2 pointed out, the investigation of the Debye length should be crucial. The authors use the Comsol Multiphysics to simulate distribution of Debye length. The results are given in Fig. RR-3c. As shown in Fig. RR-3c, the Debye length reaches its maximum at the sample position (~ 0.5 m). This value is much higher than the diameter of our reaction tube (0.042 m). It indicates that the quasi-neutrality at the sample position is destroyed. This interprets why the electron density is 2 orders lower than the ion density at the sample position in our former simulation. Moreover, the Debye length reached its maximum at the sample position of the heated furnace in 350 °C. The result is reasonable, as Debye length normally depends on density of ion number and

environmental temperature, and therefore, it increases when ion number density decreases or when the environmental temperature increases.

We also investigate the electric potential distribution in the reaction tube by COMSOL. The simulation results are shown in Fig. RR-4. As there is a huge change in the magnitude of electric potential between the plasma core region and the sample area, we plot the electric distribution in these two areas separately in different scale for clarity. As shown in Fig. RR-4, there is huge difference in the electric potential distribution. It is obvious that the electric potential in Fig. RR-4b varies more dramatically than that in Fig. RR-4a. Thus, it seems plausible that the ions around the plasma core area may obtain more energy to collectively go through the *h*-BN mesh for tremendous bubble formation. However, the ion density at plasma core area is actually 7-8 orders in magnitude higher than that at the sample position. In addition, mean free path of ions (λ) is inverse ratio to the number density in the collision process. As a result, the high density of ions may remarkably increase their collision probability with other ions and electrons, and then increase the probability of their recombination to hydrogen molecules. This could markedly shorten the mean free path of ions at the plasma core area and restrict the ions to obtain enough energy for their accelerations. As such, the density of bubbles in the plasma core area should be very high and exhibits relatively small size. Comparatively, ions can get more energy at the sample area due to the lower ion number density and finally give rise to thin but relatively large bubbles.

Figure RR-3 | Detailed distribution of **a**, Electron density; **b**, Ion density; **c**, Debye length at the sample position when the furnace is kept at 350 °C.

Figure RR-4 | Distribution of electric potential in the reaction tube. a, Details of the electric potential distribution at sample position. **b,** Details of the electric potential distribution at plasma core area.

Comment 4:

“It looks the referee thought that our setup includes an ion source and a dc electric field accelerator. Actually, our experiment setup is just a simple tube furnace equipped with a ICP generator. However, it does not affect our trial in estimation on kinetic energy of H^+ ions. “The reviewer is aware that this is not an ion source and have experience with ICP plasmas.

Reply: As the authors had limited knowledge on plasma physics, they did not fully understand the questions raised by reviewer #2 at the beginning. We apologize for our carelessness and unintentional offence. We appreciate the reviewer’s comments regarding mechanism of bubble formation, which really help us improve our manuscript.

Comment 5:

The sentence below is unclear:

“Both the duration and temperature of plasma treatment have prominent effects on the density and sizes of bubbles, which provide a feasible way to fabricate h-BN bubbles with controllable dimensions.” What means “temperature of plasma treatment” ?

Reply: Thank you for the comment.

What we want to say is the sample temperature or environmental temperature at the sample position. In the context of plasma physics, “temperature” has its special meaning, such as the “electron temperature”. The definition of “temperature” here represents the environmental temperature which is controlled by the furnace from room temperature to 350 °C.

Changes:

We have replaced all “temperature of plasma treatment” with “sample temperature” or “environmental temperature”.

Comment 6:

The main mechanisms behind the bubble formation are not clarified in the article. This is obviously plasma driven effect. The heating applied just results in some redistribution and size changing. The authors should focus on plasma mechanisms in order to give reliable explanation of the phenomenon considered. For example, changing the position of the sample along the plasma afterglow is one possible shoot to illuminate the phenomenon and to reveal the role of accelerated ions.

Reply: Thanks for the valuable suggestion.

We carried out the experiments accordingly. We placed six *h*-BN samples at Position 1-6 along in the tube as shown in Fig. RR-5 in order to investigate the difference in bubble formation. There are detailed experimental parameters: RF power in 100 W, the center of the furnace at 350 °C, a H₂ flow in a rate of 3 sccm, a pressure in 3 Pa and a duration of 60 minute for plasma treatment. AFM and optical images of the *h*-BN samples were taken and the detailed results are shown in Fig. RR-6. As shown in Fig. RR-6, the bubbles on *h*-BN sample at Position 2 (furnace center, 350 °C) and Position 6 (at the front of the RF coil) are much larger than those on other samples while there is almost no big bubble visible on *h*-BN placed at Position 5 (plasma core area near the center of the RF coil) under optical microscope. However, it is found that there are masses of tiny bubbles in a very high density on *h*-BN samples placed at Position 5 by AFM scanning. To understand these phenomena, we carried out the distribution simulation of both the electric potential (Fig. RR-4) and the ion number density (Fig. RR-7) in the tube. At the center of the RF coil (Position 5), the electric potential reaches its maximum but its gradient is much lower than that of the entrance area of RF coil (Position 6). This difference could lead to the distinction of the ion injection quantity on these two samples (the height of bubbles on *h*-BN at Position 5 is averagely < 5 nm. But for sample placed at Position 6, that is > 15 nm). Moreover, the plasma core region at Position 5 also possesses the maximal ion number density, according to Fig. RR-7. And therefore, it is reasonable that the *h*-BN flakes at Position 5 has the highest density of the bubbles among all *h*-BN samples. On the other hand, *h*-BN samples located at Position 1-4 also show different distributions of the bubble formation. At the center of the furnace, both the height and diameter values of bubbles reach their maximum of ~18 nm and ~600 nm at Position 2 while bubbles on *h*-BN at Position 1,3 and 4 could only be raised to ~7-10 nm high with the average diameter of ~100-300 nm. It is likely that the environmental temperature of 350 °C at the center of the furnace offers some additional energy to the ions, which makes them easier to penetrate the *h*-BN mesh and gives rise to the redistribution and size changing of larger bubble formation at Position 2.

Generally, the energy of the accelerated ions is determined by the comprehensive influence from ion density, Debye length and potential gradient in plasma. According to the analysis in Reply to Comment 3, the higher density of ions may remarkably increase their collision probability with other ions and electrons, and then increase the probability of their recombination to hydrogen molecules. It could markedly shorten the mean free path of ions at the plasma core area and restrict the ions to obtain enough energy for their accelerations. As such, the density of bubbles in the plasma core area should be very high and exhibits relatively small size. Comparatively, ions can get more energy at the sample area due to the lower ion number density and finally give rise to thin but relatively large bubbles. As shown in Fig. RR-6, the experimental results are consistent with our analysis in Reply to Comment 3. Bubbles in the **plasma core area** are small but dense, while bubbles **at other area** show relatively large but thin.

Figure RR-5 | Schematic of different positions to place the *h*-BN samples along the tube. (from Position 1 to 6)

Position-1

Position-2

Position-3

Position-4

Position-5

Position-6

Figure RR-6 | Optical image and AFM topographic image of *h*-BN samples placed from Position 1 to Position 6. For the *h*-BN samples at each position, **a, The optical image of *h*-BN sample. **b**, The cross-sectional profile which is corresponding to the red line marked in (e). **c-e**, AFM topographic images in different scan area.**

Figure RR-7 | Ion number density distribution in tubes, simulated by COMSOL Multiphysics. a, Details of the ion number density distribution at the plasma core region; **b,** The zoom-in view for the ion number density contour in plasma center area.

Comment 7:

Moreover, concerning hydrogen extraction from hydrocarbons, the explanations such as: “The difference in the dose of atomic hydrogen between the ethyne and methane plasmas could be a key factor that accounts for this discrepancy in the bubble formation phenomenon.” are unacceptable.

Reply: Thanks for the comment about our explanation on the differences between ethyne and methane plasmas.

Our previous explanation shows a little bit superficial, as we did not take the dissociation efficiency of two gases into description. There have been some reports about the dissociation efficiency of both CH₄ and C₂H₂. (Partridge H, Bauschlicher CW. The dissociation energies of CH₄ and C₂H₂ revisited. *The Journal of Chemical Physics*. **103**, 10589-10596 (1995).) As shown in this literature, the calculated dissociation energy of methane is smaller than that of ethyne. Normally, the CH₄ dissociation can be disassembled to 4 steps (CH₄→CH₃+H, CH₃→CH₂+H, CH₂→CH+H, CH→C+H) with the dissociation energy of 107.97, 109.93, 101.85 and 75.37 (kcal/mol), respectively, under the computing basis set of correlation consistent polarized valence double-zeta (cc-pvDZ). In the meanwhile, the C₂H₂ dissociation can also be divided to 4 steps (C₂H₂→C₂H+H, C₂H→C₂+H, C₂H₂→CH+CH, C₂→C+C) with the dissociation energy of 133.91, 105.65, 218.73 and 129.51 (kcal/mol), respectively, under the same computing basis set. It is obvious that the dissociation energy of CH₄ is less than that of the C₂H₂. It means that CH₄ exhibits higher dissociation efficiency than C₂H₂, and therefore much more atomic hydrogens in CH₄ plasma could be obtained than C₂H₂ under the same experimental conditions. Considering that the bubbles

are filled with hydrogen and all experiments with CH₄ and C₂H₂ are carried out under the same conditions (350 °C, 100 W, 120 min, ~3 sccm flow rate, ~3 Pa), we believe that more atomic hydrogen in CH₄ plasma could be one possible reason to bring hydrogen more efficiently into *h*-BN bubbles than the case in ethyne plasma. We will try a specific study for these phenomena in the future.

Change:

We replace the previous explanation with “The exact causes for the phenomena need comprehensive investigation in the future. At this moment, one reason, we believe, could be that the dissociation energy of methane is actually lower than that of the ethyne.³⁹ When all experiments were carried out under similar conditions, more atomic hydrogen in CH₄ plasma can be generated and then more efficiently get into *h*-BN bubbles than the case in ethyne plasma.” in the revised manuscript.

Comment 9:

The article is unbalanced by focusing mainly on material characterization and leaving unexplained and hidden the main plasma mechanisms resulting in bubbles formation. Keeping in mind the comments above, I cannot recommend the publication of the manuscript in its present form.

Reply: We would like to thank the reviewer for introducing us the way to figure out possible mechanisms behind the bubble formation on *h*-BN and pointing out that the lack of mechanism interpretation in our pervious manuscript. We try our best to understand the mechanisms behind the bubble formation on *h*-BN and revised our manuscript accordingly. We carried out the related simulation about plasma physics, experiments for bubbles formation at different positions in the reaction tube. We added corresponding analysis and discussion into the main text and supporting information accordingly.

Round-3 Feedback:

REVIEWERS' COMMENTS:

Reviewer #2 (Remarks to the Author):

The authors addressed all my comments and have provided explanation about the mechanism behind the bubble formation.

The article can be published in Nature Communications.